# An analog to digital converter controls bistable transfer competence development of a widespread bacterial integrative and conjugative element

Nicolas Carraro[1], Xavier Richard[1,2], Sandra Sulser[1], François Delavat[1,3], Christian Mazza[2], Jan Roelof van der Meer[1]*

[1]Department of Fundamental Microbiology, University of Lausanne, Lausanne, Switzerland; [2]Department of Mathematics, University of Fribourg, Fribourg, Switzerland; [3]UMR CNRS 6286 UFIP, University of Nantes, Nantes, France

**Abstract** Conjugative transfer of the integrative and conjugative element ICE*clc* in *Pseudomonas* requires development of a transfer competence state in stationary phase, which arises only in 3–5% of individual cells. The mechanisms controlling this bistable switch between non-active and transfer competent cells have long remained enigmatic. Using a variety of genetic tools and epistasis experiments in *P. putida*, we uncovered an 'upstream' cascade of three consecutive transcription factor-nodes, which controls transfer competence initiation. One of the uncovered transcription factors (named BisR) is representative for a new regulator family. Initiation activates a feedback loop, controlled by a second hitherto unrecognized heteromeric transcription factor named BisDC. Stochastic modelling and experimental data demonstrated the feedback loop to act as a scalable converter of unimodal (population-wide or 'analog') input to bistable (subpopulation-specific or 'digital') output. The feedback loop further enables prolonged production of BisDC, which ensures expression of the 'downstream' functions mediating ICE transfer competence in activated cells. Phylogenetic analyses showed that the ICE*clc* regulatory constellation with BisR and BisDC is widespread among *Gamma-* and *Beta*-proteobacteria, including various pathogenic strains, highlighting its evolutionary conservation and prime importance to control the behaviour of this wide family of conjugative elements.

*For correspondence:
JanRoelof.VanDerMeer@unil.ch

Competing interests: The authors declare that no competing interests exist.

## Introduction

Biological bistability refers to the existence of two mutually exclusive stable states within a population of genetically identical individuals, leading to two distinct phenotypes or developmental programs (*Shu et al., 2011*). The basis for bistability lies in a stochastic regulatory decision resulting in cells following one of two possible specific genetic programs that determine their phenotypic differentiation (*Norman et al., 2015*). Bistability has been considered as a bet-hedging strategy leading to an increased fitness of the genotype by ensuring survival of one of both phenotypes depending on environmental conditions (*Veening et al., 2008*). A number of bistable differentiation programs is well known in microbiology, notably competence formation and sporulation in *Bacillus subtilis* (*Xi et al., 2013*; *Schultz et al., 2007*), colicin production and persistence in *Escherichia coli* (*Lewis, 2007*), virulence development of *Acinetobacter baumannii* (*Chin et al., 2018*), or the lysogenic/lytic switch of phage lambda (*Sepúlveda et al., 2016*; *Arkin et al., 1998*).

Bistability may also be pervasive among many bacterial DNA conjugative systems, leading to the formation of specific conjugating donor cells at low frequency in the population (*Delavat et al., 2017*). The best described case of this is the dual lifestyle of the *Pseudomonas* integrative and

**eLife digest** Mobile DNA elements are pieces of genetic material that can jump from one bacterium to another, and even across species. They are often useful to their host, for example carrying genes that allow bacteria to resist antibiotics.

One example of bacterial mobile DNA is the ICE*clc* element. Usually, ICE*clc* sits passively within the bacterium's own DNA, but in a small number of cells, it takes over, hijacking its host to multiply and to get transferred to other bacteria. Cells that can pass on the elements cannot divide, and so this ability is ultimately harmful to individual bacteria. Carrying ICE*clc* can therefore be positive for a bacterium but passing it on is not in the cell's best interest. On the other hand, mobile DNAs like ICE*clc* have evolved to be disseminated as efficiently as possible. To shed more light on this tense relationship, Carraro et al. set out to identify the molecular mechanisms ICE*clc* deploys to control its host.

Experiments using mutant bacteria revealed that for ICE*clc* to successfully take over the cell, a number of proteins needed to be produced in the correct order. In particular, a protein called BisDC triggers a mechanism to make more of itself, creating a self-reinforcing 'feedback loop'.

Mathematical simulations of the feedback loop showed that it could result in two potential outcomes for the cell. In most of the 'virtual cells', ICE*clc* ultimately remained passive; however, in a few, ICE*clc* managed to take over its hosts. In this case, the feedback loop ensured that there was always enough BisDC to maintain ICE*clc*'s control over the cell. Further analyses suggested that this feedback mechanism is also common in many other mobile DNA elements, including some that help bacteria to resist drugs.

These results are an important contribution to understand how mobile DNAs manipulate their bacterial host in order to propagate and disperse. In the future, this knowledge could help develop new strategies to combat the spread of antibiotic resistance.

conjugative element (ICE) ICE*clc* (*Figure 1A*; *Minoia et al., 2008*). In the majority of cells ICE*clc* is maintained in the integrated state, but a small proportion of cells (3–5%) in stationary phase activates the ICE transfer competence program (*Minoia et al., 2008*; *Delavat et al., 2016*). Upon resuming growth, transfer competent (tc) donor cells excise and replicate the ICE (*Delavat et al., 2019*), which can conjugate to a recipient cell, where the ICE can integrate (*Delavat et al., 2016*). ICE*clc* transfer competence comprises a differentiated stable state, because initiated tc cells do not transform back to the ICE-quiescent state. Although tc cells divide a few times, their division is compromised by the ICE and eventually arrests completely (*Takano et al., 2019*; *Reinhard et al., 2013*).

ICEs have attracted wide general interest because of the large variety of adaptive functions they can confer to their host, including resistance to multiple antibiotics (*Waldor et al., 1996*; *Johnson and Grossman, 2015*; *Burrus et al., 2002*), or metabolism of xenobiotic compounds, such as encoded by ICE*clc* (*Miyazaki et al., 2015*; *Zamarro et al., 2016*). ICE*clc* stands model for a ubiquitous family of genomic islands found by bacterial genome sequencing, occurring in important opportunistic pathogens such as *Pseudomonas aeruginosa*, *Bordetella bronchiseptica*, *Xylella fastidiosa* or *Xanthomonas campestris* (*Miyazaki et al., 2015*). The ICE*clc* family of elements is characterized by a consistent 'core' region of some 50 kb (*Figure 1A*), predicted to encode conjugative functions, and a highly diverse set of variable genes with adaptive benefit (*Miyazaki et al., 2015*). Strong core similarities between ICE*clc* and the PAGI-2 family of pathogenicity islands in *P. aeruginosa* clinical isolates have been noted previously (*Miyazaki, 2011a*; *Klockgether et al., 2007*).

Although the existence and the fitness consequences of the ICE*clc* bistable transfer competence pathway have been studied in quite some detail, the regulatory basis for its activation has remained largely elusive (*Delavat et al., 2017*). In terms of its genetic makeup, ICE*clc* seems very distinct from the well-known SXT/R391 family of ICEs (*Wozniak and Waldor, 2010*) and from ICE*St1*/ICE*St3* of *Streptococcus thermophilus* (*Carraro and Burrus, 2014*). These carry analogous genetic regulatory circuitry to the lambda prophage, which is characterized by a typical double-negative feedback control (*Poulin-Laprade and Burrus, 2015*; *Bellanger et al., 2007*). Transcriptomic studies indicated that the core region of ICE*clc* (*Figure 1A*) is higher expressed in stationary than exponential phase cultures grown on 3-chlorobenzoate (3-CBA), and organized in at least half a dozen transcriptional

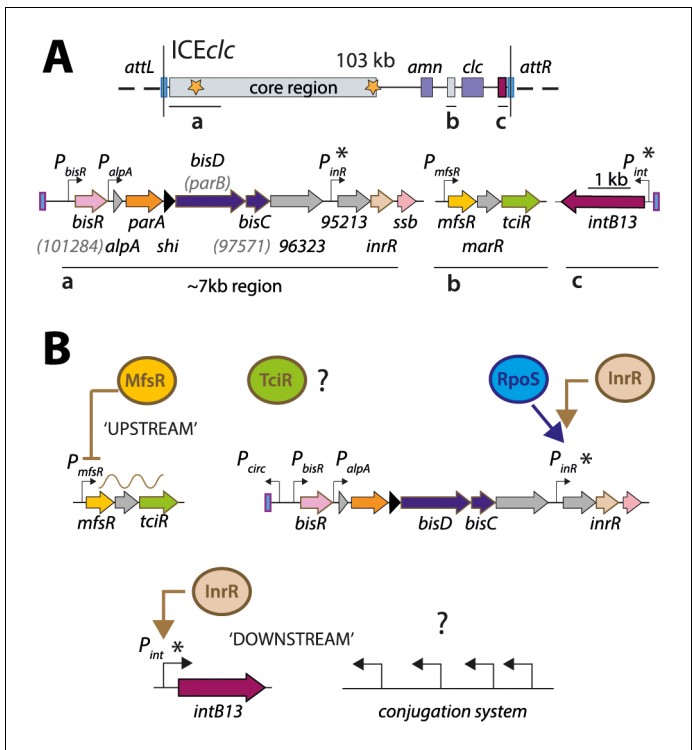

**Figure 1.** ICE*clc* and postulated regulation network for transfer competence formation. (**A**) Schematic representation of the genetic organization of ICE*clc* (GenBank accession number AJ617740.2). Loci of interest (**a, b** and **c**) are detailed below the general map and drawn to scale. Note the ~7 kb left-end region, which is the major focus of the study. Genes are represented by coloured arrows with their name below (former names shown in lighter font inside brackets). Promoters are represented by hooked arrows pointing towards the transcription orientation. Those marked with an asterisk are known to be expressed only in the subpopulation of transfer competent cells. *attL* and *attR,* attachment sites; *clc* genes: 3-chlorocatechol degradation, *amn* genes: 2-aminophenol degradation. (**B**) Known steps in ICE*clc* transfer competence regulation. An 'upstream' cascade, with MfsR autorepressing its own transcription and that of *tciR*; TciR overexpression leading to transfer competence in almost all cells (***Pradervand et al., 2014***). Bistable expression of 'downstream' genes from $P_{inR}$ and $P_{int}$ in the subpopulation of transfer competent cells, and further roles of additional factors RpoS (***Miyazaki et al., 2012***) and InrR (***Minoia et al., 2008***).

The online version of this article includes the following figure supplement(s) for figure 1:

**Figure supplement 1.** Postulated model of bistability generation of ICE*clc.*

**Figure supplement 2.** Protein domain predictions in the ICE*clc* bistability regulators BisR, BisD and BisC, and AlpA.

**Figure supplement 3.** Gene organization of ICE*clc*-related regulatory loci in different bacterial species.

**Figure supplement 4.** BisDC homologs among *Proteobacteria*.

units (***Gaillard et al., 2010***). A group of three consecutive regulatory genes precludes ICE*clc* activation in exponentially growing cells, with the first gene (*mfsR*) constituting a negative autoregulatory feedback (***Figure 1B***; ***Pradervand et al., 2014***). Overexpression of the most distal of the three genes (*tciR*), leads to a dramatic increase of the proportion of cells activating the ICE*clc* transfer competence program (***Pradervand et al., 2014***). Despite this initial discovery, however, the nature of the regulatory network architecture leading to bistability and controlling further expression of the ICE*clc* genes in tc cells has remained enigmatic.

The primary goal of this work was to dissect the regulatory factors and nodes underlying the activation of ICE*clc* transfer competence. Secondly, given that transfer competence only arises in a small proportion of cells in a population, we aimed to understand how the regulatory architecture yields and maintains ICE bistability. We essentially followed two experimental strategies and phenotypic readouts. First, known and suspected regulatory elements were seamlessly deleted from ICE*clc* in *P.*

*putida* and complemented with inducible plasmid-cloned copies to study their epistasis in transfer of the ICE. Secondly, individual and combinations of suspected regulatory elements were expressed in a *P. putida* host without ICE, to study their capacity to activate the ICE*clc* promoters $P_{int}$ and $P_{inR}$, which normally only express in wild-type tc cells (*Figure 1B*; *Minoia et al., 2008*). As readout for their activation we quantified fluorescent reporter expression from single copy chromosomally integrated transcriptional fusions, as well as the proportion of cells expressing the reporters using subpopulation statistics as previously described (*Reinhard and van der Meer, 2013*). On the basis of the discovered key regulators and nodes, we then developed a conceptual mathematical model to show by stochastic simulations how bistability is generated and maintained. This suggested that the ICE*clc* transfer competence regulatory network essentially converts a unimodal (analog) input signal from the 'upstream' regulatory branch occurring in all cells (*Figure 1B*) to a bistable (digital) output in a subset, and in scalable manner. We experimentally verified this scalable analog-digital conversion in a *P. putida* without ICE*clc* but with the reconstructed bistability generator. The key ICE*clc* bistability regulatory elements involve new previously unrecognized transcription factors, which are conserved among a wide range of Proteobacteria, illustrating their importance for the behaviour of this conglomerate of related ICEs.

## Results

### Activation of ICE*clc* starts with the LysR-type transcription regulator TciR

Previous work had implied an ICE*clc*-located operon of three consecutive regulatory genes (*mfsR*, *marR* and *tciR*, *Figure 1B*) in control of transfer competence formation (*Pradervand et al., 2014*). That work had shown that *mfsR* codes for an autorepressor, whose deletion yielded unhindered production of the LysR-type activator TciR. As a result, the proportion of tc cells is largely increased in *P. putida* UWC1 bearing ICE*clc*-Δ*mfsR* (*Delavat et al., 2016*; *Pradervand et al., 2014*). We reproduced this state of affairs here by cloning *tciR* under control of the IPTG-inducible $P_{tac}$ promoter on a plasmid (pME*tciR*) in *P. putida* UWC1-ICE*clc*. In absence of cloned *tciR*, transfer of wild-type ICE*clc* from succinate-grown *P. putida* to an ICE*clc*-free isogenic *P. putida* was below detection limit, indicating that spontaneous ICE activation under those conditions is negligible (*Figure 2A*). In contrast, inducing *tciR* expression by IPTG addition triggered ICE*clc* transfer from succinate-grown cells up to frequencies close to those observed under wild-type growth conditions with 3-CBA (*Miyazaki and van der Meer, 2011b*) ($10^{-2}$ transconjugant colony-forming units (CFU) per donor CFU, *Figure 2A*). Transfer frequencies were lower in the absence of IPTG, which indicated that leaky expression of *tciR* from $P_{tac}$ was sufficient to trigger ICE*clc* transfer (*Figure 2A*). These results confirmed the implication of TciR and thus we set out to identify its potential activation targets on ICE*clc*.

Induction of *tciR* from pME*tciR* in *P. putida* without ICE*clc* was insufficient to trigger eGFP production from a single-copy $P_{int}$ promoter, which is a hallmark of induction of ICE*clc* transfer competence (*Figure 2B*; *Minoia et al., 2008*; *Delavat et al., 2016*). In contrast, in presence of ICE*clc*, similar induction of *tciR* yielded a clear increased subpopulation of activated cells (*Figure 2C*). This suggested, therefore, that TciR does not directly activate $P_{int}$, but only through one or more other ICE-located factors. To search for such potential factors, we examined in more detail the genes in a 7 kb region at the left end of ICE*clc* (close to the *attL* site, *Figure 1A*), where transposon mutations had previously been shown to influence $P_{int}$ expression (*Sentchilo et al., 2003*). In addition, three promoters had been characterized in this region (*Figure 1A*; *Gaillard et al., 2010*), which we tested individually for potential activation by TciR (*Figure 2B*).

Promoters were fused with a promoterless *egfp* gene and inserted in single copy into the chromosome of *P. putida* UWC1 without ICE*clc* (Materials and methods). Induction of *tciR* from $P_{tac}$ on pME*tciR* did not yield any eGFP fluorescence in *P. putida* UWC1 containing a single-copy $P_{alpA}$- or $P_{inR}$-*egfp* transcriptional fusion (*Figure 2B*). In contrast, the $P_{bisR}$-*egfp* fusion was activated upon induction of TciR compared to a vector-only control (p=0.0042, paired t-test, *Figure 2B & D*). This suggested that the link between TciR and ICE*clc* transfer competence proceeds through transcription activation of the promoter upstream of the gene *bisR* (previously designated *orf101284*). This transcript has previously been mapped and covers a single gene (*Gaillard et al., 2010*). We renamed

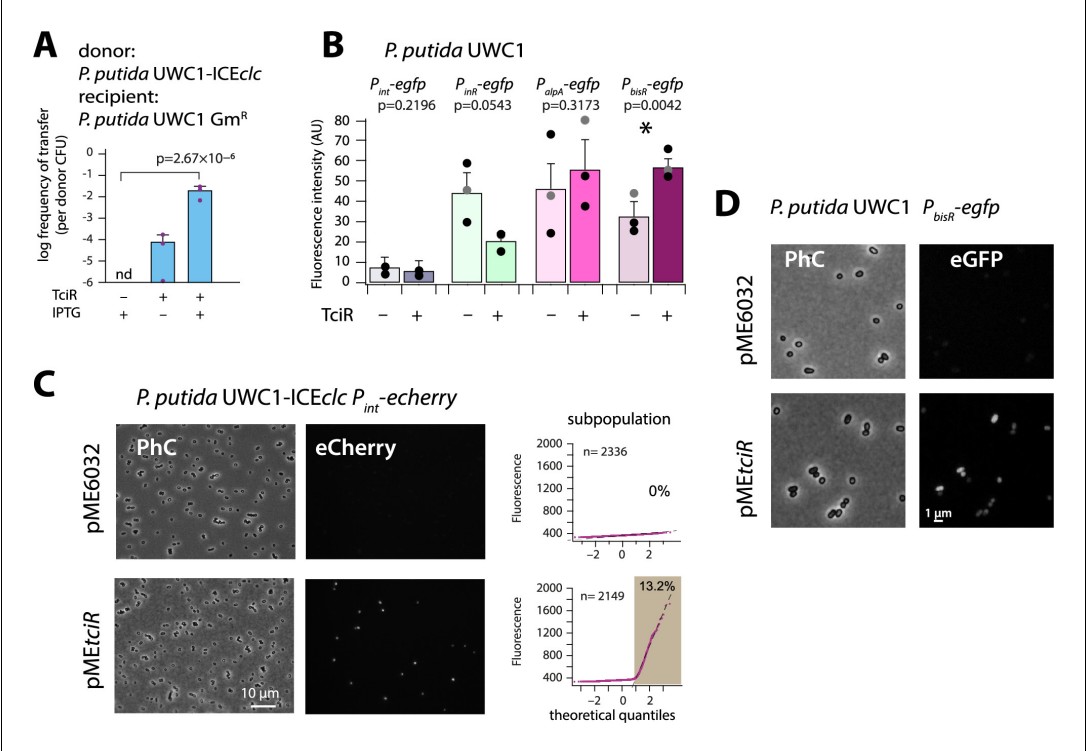

**Figure 2.** The LysR-type regulator TciR links to a single node in the regulation network. (A) Ectopic overexpression of *tciR* induces ICE*clc* wild-type conjugative transfer under non-permissive conditions. Bars show the means (+ one standard deviation) of transconjugant formation after 48 hr in triplicate matings using *P. putida* UWC1 donors carrying the indicated ICE*clc* or plasmids, in absence (-) or presence (+) of 0.1 mM IPTG, and with a Gm$^R$-derivative of *P. putida* as recipient. Dots represent individual transfer; nd: not detected (<10$^{-7}$ for the three replicates). p-value derives from one-sided t-test comparison (*n* = 3). (B) Reporter expression from single copy chromomosomal $P_{bisR}$, $P_{inR}$, $P_{int}$, or $P_{alpA}$ transcriptional *egfp* fusions in *P. putida* UWC1 without ICE*clc* as a function of ectopically expressed TciR, in comparison to strains carrying the empty vector pME6032. Bars show means of the 75th percentile fluorescence of 500–1000 individual cells each per triplicate culture grown on succinate, induced with 0.05 mM IPTG. Error bars denote standard deviation from the means from biological triplicates (dots show individual 75th percentiles). AU, arbitrary units of brightness at 500 ms exposure. p-values derive from pair-wise comparisons in t-tests between cultures expressing TciR and not. (C) Proportion of cells expressing eCherry from a single-copy chromosomal insertion of $P_{int}$ in *P. putida* with ICE*clc* in presence of induced TciR (pME*tciR*, 0.05 mM IPTG) or with empty vector (pME6032). Fluorescence images scaled to same brightness (300–2000). Diagrams show quantile-quantile plots of individual cell fluorescence levels, with *n* denoting the number of analysed cells and the shaded part indicating the subpopulation size expressing $P_{int}$-*echerry*. (D) Fluorescence images of *P. putida* without ICE*clc* with a single-copy chromosomal $P_{bisR}$-*egfp* fusion in presence of empty vector or of induced TciR. Images scaled to same brightness (300–1200).

The online version of this article includes the following source data for figure 2:

Source data 1. *Figure 2* panel A: ICE*clc* transfer data.
Source data 2. *Figure 2* panel B: 75th percentile fluorescence data.
Source data 3. *Figure 2* panel C: qq-plot single cell fluorescence data.

this gene as *bisR*, or <u>bis</u>tability <u>r</u>egulator, for its presumed implication in ICE*clc* bistability control (***Figure 1—figure supplement 1***, see further below).

## BisR is the second step in the cascade of ICE*clc* transfer competence initiation

*bisR* is predicted to encode a 251-aa protein of unknown function with no detectable Pfam-domains. Further structural analysis using Phyre2 (***Kelley et al., 2015***) suggested three putative domains with low confidence (between 38% and 53%, ***Figure 1—figure supplement 2***). One of these is a predicted DNA-binding domain, which hinted at the possible function of BisR as a transcriptional regulator itself. BlastP analysis showed that BisR homologs are widely distributed and well conserved

among *Beta-*, *Alpha-* and *Gammaproteobacteria*, with homologies ranging from 43–100% amino acid identity over the (quasi) full sequence length (*Figure 1—figure supplement 2*).

In order to investigate its potential regulatory function, *bisR* was cloned on a plasmid (pME*bisR*) and introduced into *P. putida* UWC1-ICE*clc*. Inducing *bisR* by IPTG addition from $P_{tac}$ triggered high rates of ICE*clc* transfer on succinate media (*Figure 3A*). Deletion of *bisR* on ICE*clc* abolished its transfer, even upon overexpression of *tciR*, but could be restored upon ectopic expression of *bisR* (*Figure 3A*). This showed that the absence of transfer was due to the lack of intact *bisR*, and not to a polar effect of *bisR* deletion on a downstream gene (*Figure 1A*). In addition, transfer of an ICE*clc* deleted for *tciR* (*Pradervand et al., 2014*) could be restored by ectopic *bisR* expression

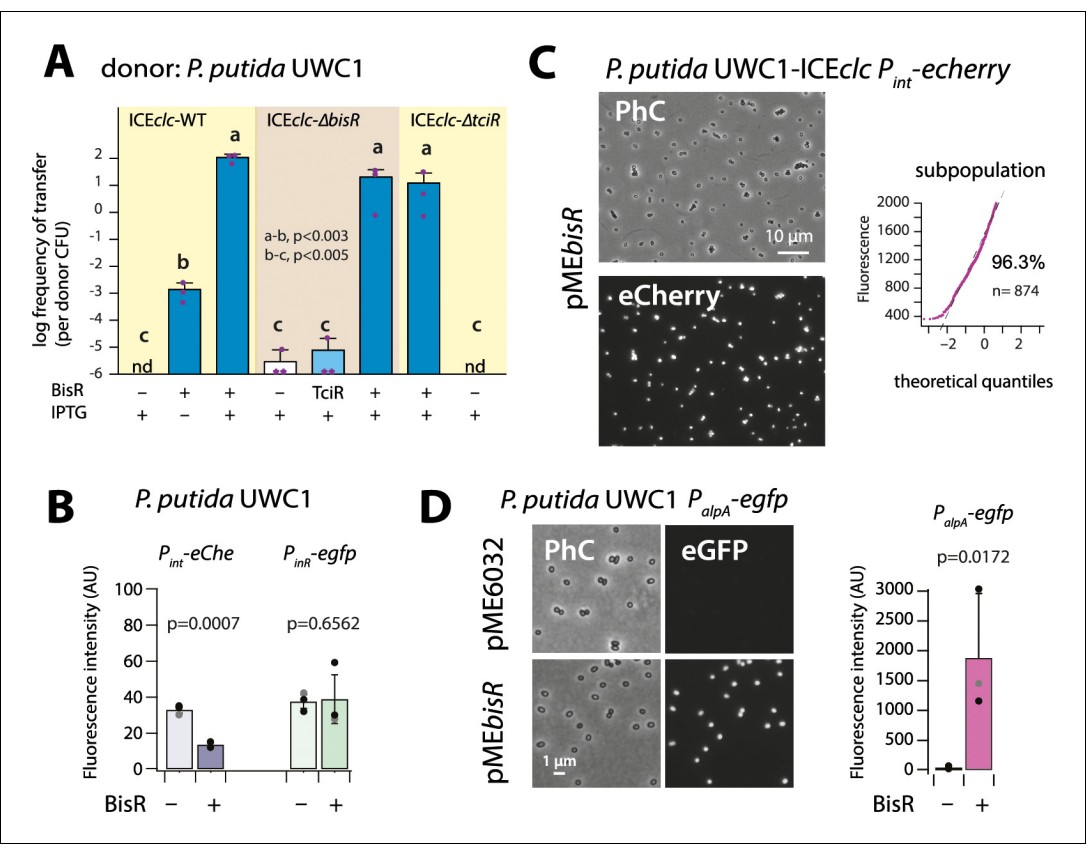

**Figure 3.** Identification of BisR as a new intermediary regulator for $P_{alpA}$ activation. (**A**) Ectopic overexpression of *bisR* induces ICE*clc* conjugative transfer under non-permissive conditions and from ICE*clc* deleted of key regulatory genes. For explanation of bar diagram meaning, see *Figure 2A* legend. BisR (+), plasmid with *bisR*; TciR, pME*tciR*; –, empty vector pME6032. nd: not detected ($<10^{-7}$ for the three replicates). Letters indicate significance groups in ANOVA followed by post-hoc Tukey testing (e.g., a-b: p-values between groups **a** and **b**; b-c: p-values between groups **b** and **c**). (**B**) Absence of direct induction by BisR of $P_{int}$ or $P_{inR}$ fluorescence reporters in *P. putida* without ICE. For explanation of bars, see *Figure 2B* legend. (**C**) Population-wide expression of $P_{int}$-*echerry* in *P. putida* with ICE*clc* upon ectopic induction of plasmid-located BisR (pME*bisR*, 0.05 mM IPTG). Image brightness scale: 300–2000. For vector control, see *Figure 2C*. (**D**) Induced BisR from plasmid leads to reporter expression from the *alpA*-promoter in all cells of *P. putida* without ICE*clc*. Image brightness scales: 300–1200. Bars show means and standard deviation from median fluorescence intensity of single cells (*n* = 500–1000, summed from 6 to 12 images per replicate) of biological triplicates. p-value derives from pair-wise t-test between cultures with empty vector (–) and those with induced BisR (+).

The online version of this article includes the following source data for figure 3:

**Source data 1.** *Figure 3* panel A: ICE*clc* transfer frequencies.
**Source data 2.** *Figure 3* panel C: qq plot single cell fluorescence values.
**Source data 3.** *Figure 3* panel B and panel D: 75th percentile fluorescence data.

(*Figure 3A*). This indicated that TciR is 'upstream' in the regulatory cascade of BisR, and that TciR does not act anywhere else on the expression of components crucial for ICE*clc* transfer.

IPTG induction of *bisR* in *P. putida* without ICE again did not yield activation of the single-copy $P_{int}$ or $P_{inR}$ transcriptional reporter fusions, whereas some repression was observed on $P_{int}$ itself (*Figure 3B*). In contrast, BisR induction in *P. putida* UWC1 with ICE*clc* led to a massive activation of the same reporter constructs in virtually all cells (*Figure 3C*), compared to a vector-only control (*Figure 2C*, pME6032). This suggested that BisR was an(other) intermediate regulator step in the complete cascade of activation of ICE*clc* transfer competence. Of the tested ICE–promoters within this 7 kb region, BisR induction triggered very strong expression from a single copy $P_{alpA}$–*egfp* transcriptional fusion in all cells (*Figure 3D*). This indicated that BisR is a transcription activator, and an intermediate regulator between TciR and further factors encoded downstream of the *alpA*-promoter (*Figure 1—figure supplement 1*).

## A new regulator BisDC is the last step in the activation cascade

Next, we thus focused our attention on the genes downstream of the *alpA*-promoter. Cloning the genes from *alpA* all the way to *inrR* (*Figure 1A*) on plasmid pME6032 under control of $P_{tac}$ and inducing that construct with IPTG resulted in activation of $P_{inR}$–*egfp* and $P_{int}$–*echerry* expression in *P. putida* without ICE*clc* (*Figure 4A*). Both these promoters had been silent upon activation of TciR or BisR (*Figure 2B* and *Figure 3B*). This indicated that one or more regulatory factors directly controlling expression of $P_{inR}$ and/or $P_{int}$ were encoded in this region, which we tried to identify by subcloning different gene configurations.

Removing *alpA* from the initial construct had no measurable effect on expression of the fluorescent reporters, but replacing $P_{tac}$ by the native $P_{alpA}$ promoter abolished all $P_{int}$ reporter activation (*Figure 4B*, *Figure 4—figure supplement 1*). This suggested that $P_{alpA}$ is silent without activation by BisR (see below) and no spontaneous production of regulatory factors occurred. Removing three genes at the 3' extremity (i.e., *orf96323*, *orf95213* and *inrR*) reduced $P_{int}$–*echerry* reporter expression, but a fragment with a further deletion into the *bisC* gene was unable to activate $P_{int}$ (*Figure 4B*). Induction of *inrR* alone did not result in $P_{int}$ activation (*Figure 4B*). Deletion of *parA* and *shi* at the 5' end of the fragment still enabled reporter expression from $P_{int}$, narrowing the activator factor regions down to two genes, previously named *parB* and *orf97571*, but renamed here to *bisD* and *bisC* (*Figure 4B*). Neither *bisC* or *bisD* alone, but only the combination of *bisDC* resulted in reporter expression from $P_{int}$ in *P. putida* UWC1 without ICE*clc* (*Figure 4B*), and similarly, of $P_{inR}$ (*Figure 4—figure supplement 1*). In the presence of ICE*clc*, inducing either *bisC* or *bisD* from a plasmid yielded a small proportion of cells expressing the $P_{int}$ reporters (*Figure 4C*). This was not the case in a *P. putida* carrying an ICE*clc* with a deletion of *bisD* (*Figure 4—figure supplement 2*), suggesting there was some sort of feedback mechanism of BisDC on itself (see further below). In contrast, induction of *bisDC* in combination caused a majority of cells to express fluorescence from $P_{int}$ in *P. putida* containing ICE*clc* (*Figure 4C*) or ICE*clc*-Δ*bisD* (*Figure 4—figure supplement 2*). These results indicated that BisDC acts as an ensemble to activate transcription, and this pointed to *bisDC* as the last step in the regulatory cascade, since it was the minimum unit sufficient for activation of the $P_{int}$–promoter, which is exclusively expressed in the subpopulation of tc cells of wild-type *P. putida* with ICE*clc* (*Delavat et al., 2016*; *Figure 1—figure supplement 1*).

Induction of *bisDC* from plasmid pME*bisDC* yielded high frequencies of ICE*clc* transfer from *P. putida* UWC1 under succinate-growth conditions (*Figure 4D*). Expression of BisDC also induced transfer of ICE*clc*-variants deleted for *tciR* or for *bisR* (*Figure 4D*). This confirmed that both *tciR* and *bisR* relay activation steps to $P_{bisR}$ and $P_{alpA}$, respectively, but not to further downstream ICE promoters (*Figure 1—figure supplement 1*). Moreover, an ICE*clc* deleted for *bisD* could not be restored for transfer by overexpression of *tciR* or *bisR*, but only by complementation with *bisDC* (*Figure 4D*). Interestingly, the frequency of transfer of an ICE*clc* lacking *bisD* complemented by expression of *bisDC* in trans was two orders of magnitude lower than that of similarly complemented wild-type ICE*clc*, ICE*clc* with *tciR*- or *bisR*-deletion (*Figure 4D*). This was similar as the reduction in reporter expression observed in *P. putida* ICE*clc*-Δ*bisD* complemented with pME*bisDC* compared to wild-type ICE*clc* (*Figure 4—figure supplement 2*), and suggested the necessity of some 'reinforcement' occurring in the wild-type configuration that was lacking in the *bisD* deletion and could not be restored by in trans induction of plasmid-cloned *bisDC*.

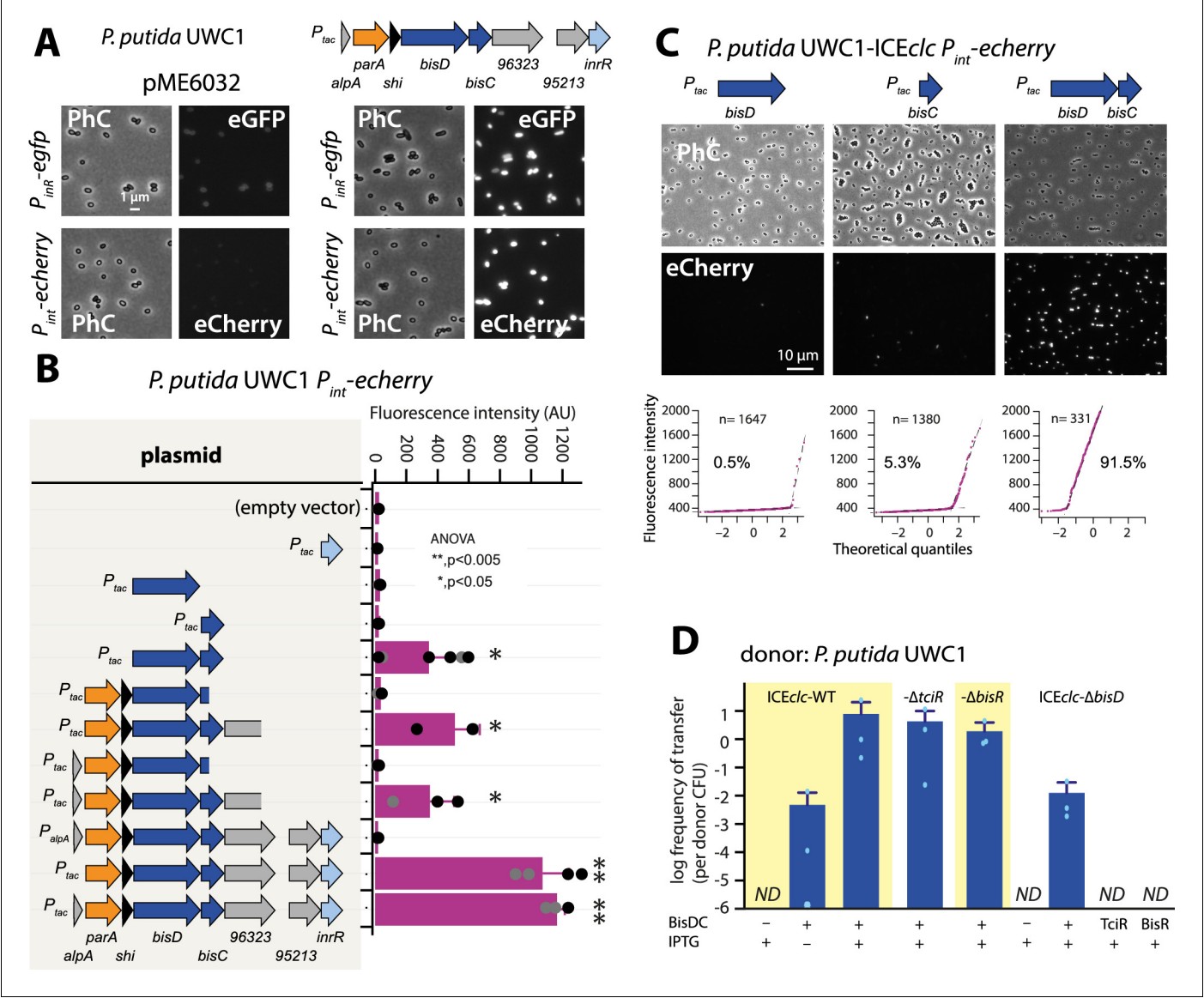

**Figure 4.** A new regulatory factor BisDC for activation of downstream ICE*clc* functions. (A) IPTG (0.05 mM) induction of a plasmid with the cloned ICE*clc* left-end gene region (as depicted on top) leads to reporter expression from the 'downstream' $P_{inR}$- and $P_{int}$-promoters in *P. putida* without ICE*clc*. Fluorescence images scaled to same brightness (300–1200). (B) $P_{int}$-*echerry* reporter expression upon IPTG induction (0.05 mM) of different plasmid-subcloned left-end region fragments (grey shaded area on the left) in *P. putida* without ICE*clc*. Bars show means of median cell fluorescence levels with one standard deviation, from triplicate biological cultures (n = 500–1000 cells, summed from 6 to 12 images per replicate). Asterisks denote significance groups in ANOVA followed by post-hoc Tukey testing. (C) Population response of $P_{int}$-*echerry* induction in *P. putida* with ICE*clc* in presence of plasmid constructs expressing *bisD*, *bisC* or both (fluorescence images scaled to 300–2000 brightness). Quantile-quantile plots (n = number of cells) below show the estimated size of the responding subpopulation. (D) Effect of *bisDC* induction from cloned plasmid (0.05 mM IPTG) on conjugative transfer of ICE*clc* wild-type or mutant derivatives. Transfer assays as in legend to *Figure 2A*. *ND*, below detection limit ($10^{-7}$).

The online version of this article includes the following source data and figure supplement(s) for figure 4:

**Source data 1.** *Figure 4* panel B: median fluorescence values.
**Source data 2.** *Figure 4* panel C: qq plot single cell fluorescence data.
**Source data 3.** *Figure 4* panel D: ICE*clc* transfer data.
**Figure supplement 1.** Effect of *bisDC* containing plasmid constructs on fluorescent protein expression from a single copy dual $P_{int}$-$P_{inR}$ reporter in *P. putida* without ICE*clc*.
**Figure supplement 2.** Effect of *bisDC* expression on reporter expression from $P_{int}$ in *P. putida* carrying ICE*clc* with a *bisD* deletion.

## BisDC is part of a positive autoregulatory feedback loop

To investigate this potential 'reinforcement' in wild-type configuration, we revisited the potential for activation of the *alpA* promoter. Induction by IPTG of the plasmid-cloned fragment encompassing the gene region *parA-shi-bisDC* caused strong activation of reporter gene expression from $P_{alpA}$ in *P. putida* without ICE*clc* (*Figure 5A*). The minimal region that still maintained $P_{alpA}$ induction encompassed *bisDC*, although much lower than with a cloned *parA-shi-bisDC* fragment (*Figure 5A*). Interestingly, when the *parA-shi-bisDC* fragment was extended by *alpA* itself, reporter expression from $P_{alpA}$ was abolished, whereas also a fragment containing only *alpA* caused significant repression of the *alpA* promoter (*Figure 5A*). The *alpA* gene is predicted to encode a 70-amino acid DNA binding protein with homology to phage regulators (*Trempy et al., 1994*; *Figure 1—figure supplement 2*). These results would imply feedback control on activation of $P_{alpA}$, since its previously mapped transcript covers the complete region from *alpA* to *orf96323* on ICE*clc*, including *bisDC* (*Figure 1A*; *Gaillard et al., 2010*). Although induction of BisDC was sufficient for activation of transcription from $P_{alpA}$, this effectively only yielded a small subpopulation of cells with high reporter fluorescence values (*Figure 5B & C*), in contrast to induction of the larger cloned gene region encompassing *parA-shi-bisDC* that activated all cells (*Figure 5B & C*). The feedback loop, therefore, seemed to consist of a positive forward part that includes BisDC (reinforced by an as yet unknown other mechanism) and a modulatory repressive branch including AlpA (*Figure 1—figure supplement 1*).

## Modelling suggests positive feedback loop to generate and maintain ICE*clc* bistable output

The results so far thus indicated that ICE*clc* transfer competence is initiated by TciR activating transcription of the promoter upstream of *bisR*. BisR then kickstarts expression from the *alpA*-promoter, leading to (among others) expression of BisDC. This is sufficient to induce the 'downstream' ICE*clc* transfer competence pathway (*Figure 1—figure supplement 1*), exemplified here by activation of the $P_{int}$ and $P_{inR}$ promoters that become exclusively expressed in the subpopulation of transfer competent cells under wild-type conditions (*Minoia et al., 2008*). In addition, BisDC reinforces transcription from the same *alpA*-promoter.

In order to understand the importance of this regulatory architecture for generating bistability, for initiating and maintaining (downstream) transfer competence, we developed a conceptual mathematical model (*Figure 6A*, Materials and methods, SI model). The model assumes the regulatory factors TciR, BisR and BisDC, typical oligomerization (*Tropel and van der Meer, 2004*), as well as binding of the oligomerized forms to and unbinding from their respective nodes (i.e., the linked promoters $P_{bisR}$, $P_{alpA}$ and $P_{int}$). Binding is assumed to lead to protein synthesis and finally, protein degradation (*Figure 6A*). We varied and explored the outcomes of different regulatory network architectures and parts, testing their effect on production of intermediary and downstream elements in stochastic simulations, with each individual simulation corresponding to events taking place in an individual cell (*Figure 6A,SI* model).

First we simulated the cellular output of BisDC in a subnetwork configuration with only BisDC activating $P_{alpA}$ (i.e., in absence of TciR or BisR, *Figure 6B*). Stochastic simulations (n = 10,000) of this bare feedback loop with an arbitrary start of binomially distributed BisDC quantities (mean = 8 molecules per cell, *Figure 6B*, *INPUT*), yielded a bimodal population with two BisDC output states after 100 time steps, one of which is *zero* (black bar in histograms) and the other with a mean *positive* BisDC value (magenta) (*Figure 6B*). The output *zero* results when BisDC levels stochastically fall to 0 (as in case of the light blue line in the panel *STOCHASTIC* of *Figure 6B*), since in that case there is no BisDC to stimulate its own production. Parameter variation showed that the proportion of cells with output *zero* from the loop is dependent on the binding and unbinding constants of BisDC to the *alpA* promoter, and the BisDC degradation rate (*Figure 6B*, different *A1*, *A2* and *A4*-values). In addition, BisDC unbinding and degradation rates can influence the median BisDC output quantity in cells with *positive* state (*Figure 6B*, case of *A2 = 5* or *A4 = 0.3*). This simulation thus indicated that a BisDC feedback loop can produce bimodal output, once BisDC is present.

Since the feedback loop cannot start without BisDC, it is imperative to kickstart the *alpA* promoter by BisR (*Figure 6C*). Simulations of a configuration that includes activation by BisR, showed how upon a single pulse of BisR, the feedback loop again leads to a bimodal population with *zero*

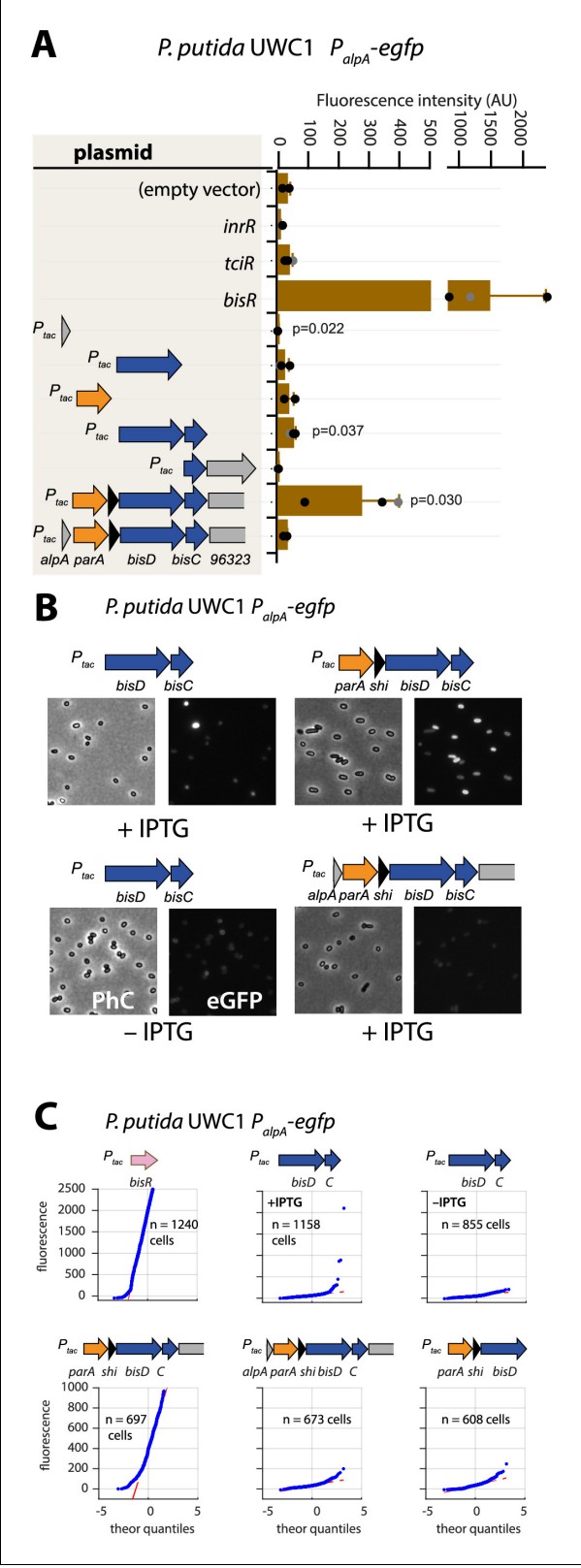

**Figure 5.** Autoregulatory feedbacks on the *alpA* promoter. (**A**) $P_{alpA}$-reporter expression upon IPTG induction (0.05 mM) of plasmid-cloned individual genes or gene combinations (as depicted in the shaded area on the left) in *P. putida* without ICE*clc*. Bars represent means of median cell fluorescence plus one standard deviation, as in legend to *Figure 2B*. p-values stem from pair-wise comparisons between triplicate cultures carrying the empty

*Figure 5 continued on next page*

*Figure 5 continued*

vector pME6032 and the indicated plasmid-cloned gene(s). (**B**) Cell images of *P. putida* $P_{alpA}$-*egfp* without ICE*clc* expressing plasmid-cloned combinations with *bisDC* (fluorescence brightness scaled to 300–1200, 0.05 mM IPTG). (**C**) Quantile-quantile estimation of subpopulation expression of the $P_{alpA}$-*egfp* reporter, showing the sufficiency of *bisDC* induction for autoregulatory feedback and the reinforcement from upstream elements (*n* denotes the number of cells used for the quantile-quantile plot, summed from 6 to 12 images of a single replicate culture). The online version of this article includes the following source data for figure 5:

**Source data 1.** *Figure 5* panel A: Median fluorescence values.
**Source data 2.** *Figure 5* panel C : qq plot single cell fluorescence values.

and *positive* BisDC levels (*Figure 6C*). Increasing the (uniformly distributed) mean quantity of BisR in the simulations, within a per-cell range that is typically measured for transcription factors (*Li et al., 2014*), increased the proportion of cells with *positive* BisDC state, but did not influence their mean BisDC quantity (*Figure 6C*). Even bimodally distributed BisR input also gave rise to bimodal BisCD output, but with a higher proportion of *zero* BisDC state (*Figure 6C*, bimodal). In contrast to the BisDC loop alone, therefore, activation by BisR only influences the proportion of *zero* and *positive* BisDC states in the population, but not the mean resulting BisDC quantity in cells with *positive* state.

In the full regulatory hierarchy of the ICE, production of BisR is controlled by TciR. Simulation of this configuration showed that bimodality already appeared at the level of BisR (*Figure 6D*). The proportions of *zero* and *positive* states of both BisR and BisDC varied depending on the mean of uniformly distributed amounts of TciR among all cells, again sampled to within regular empiric transcription regulator quantities in individual cells (*Li et al., 2014*; *Figure 6D*). Bimodal BisDC levels are propagated by the network architecture to downstream ('late') promoters, as a consequence of them being under BisDC control (*Figure 6A & E*). Importantly, simulations of an architecture without the BisDC feedback loop consistently resulted in lower protein output from BisDC–regulated promoters in activated cells than with feedback (*Figure 6E*). This suggests two crucial functions for the ICE regulatory network: first, to convert unimodal or stochastic ('analog') expression of TciR and BisR among all cells to a consistent subpopulation of cells with *positive* ('digital') BisDC state, and secondly, to ensure sufficient BisDC levels to activate downstream promoters within the *positive* cell population (*Figure 6E*). Through the kickstart by BisR and reinforcement by BisDC itself, bimodal expression at the *alpA*-promoter node can thus yield a stably expressed transfer competence pathway in a subpopulation of cells.

## ICE*clc* regulatory architecture exemplifies a faithful analog-to-digital converter

Simulations thus predicted that the ICE regulatory network faithfully transmits and stabilizes analog input (e.g., a single regulatory factor uniformly or stochastically expressed at moderately low levels in all cells [*Li et al., 2014*]) to bistable output (e.g., a subset of cells with transfer competence and the remainder silent). To demonstrate this experimentally, we engineered a *P. putida* without ICE*clc*, but with a single copy chromosomally inserted IPTG-inducible *bisR*, a plasmid with *alpA-parA-shi-bisDC* under control of $P_{alpA}$, and a single-copy dual $P_{int}$-*echerry* and $P_{inR}$-*egfp* reporter (*Figure 7A*). Induction from $P_{tac}$ by IPTG addition yields unimodal (analog) production of BisR, the mean level of which can be controlled by the IPTG concentration (*Figure 7—figure supplement 1*). In the presence of all components of the system, IPTG induction of BisR led to bistable activation of both reporters (*Figure 7B*, *ABC*). Increasing BisR induction was converted by the feedback loop into an increased proportion of fluorescent cells (*Figure 7C*). This effectively created a scalable bimodal (digital) output from unimodal input, dependent on the used IPTG concentration (*Figure 7C*, *Figure 7—figure supplement 1*). The proportion of fluorescent cells was in line with predictions from stochastic simulations as a function of the relative strength of $P_{tac}$ activation (*Figure 7D*). Furthermore, in agreement with model predictions (*Figure 6C*), the median fluorescence of activated cells remained the same at different IPTG (and thus BisR) concentrations (*Figure 7E*). These results confirmed that the feedback loop architecture transforms a unimodal (analog) regulatory factor concentration (BisR) into a stabilized bimodal (digital) output.

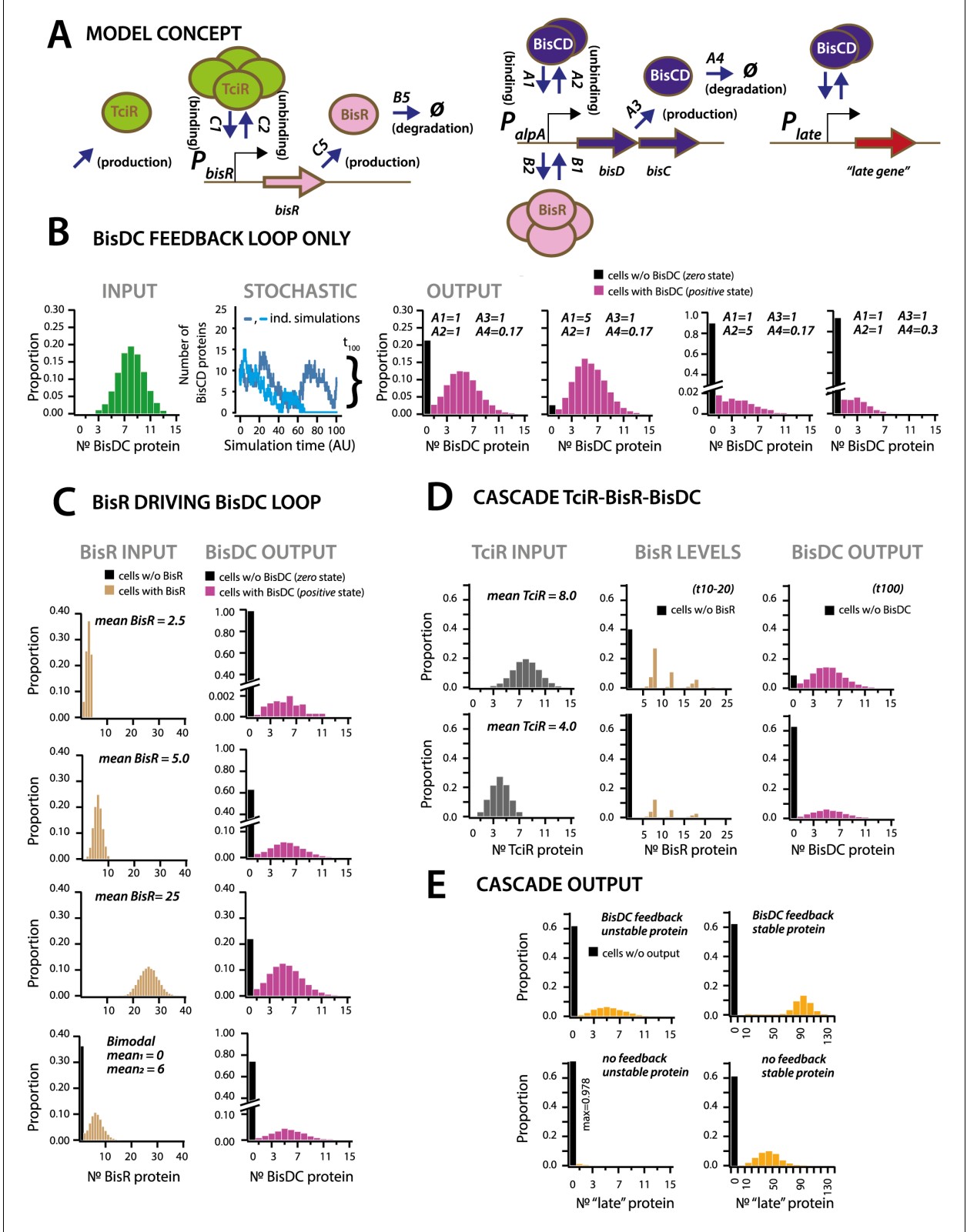

**Figure 6.** Stochastic simulations of ICE*clc* regulatory network configurations. (**A**) Conceptual model of the ICE*clc* regulatory cascade producing bistable output. Ellipses indicate the three major regulatory factors (TciR, BisR and BisDC) interacting with their target promoters ($P_{bisR}$, $P_{alpA}$), and BisDC-regulated downstream output (here schematically as $P_{late}$ and a *late* gene). Relevant simulated processes include: *production* (combination of transcription and translation, with corresponding rates: C5, A3), *oligomerization* (assumed number of protein monomers in the binding complex),

Figure 6 continued

binding and unbinding to the target promoter, and degradation. All processes are simulated as stochastic events across 100 time steps, and protein output levels are summarized from 10,000 individual stochastic simulations (curly bracket in B; detailed parametrization in *Supplementary file 4*; one simulation being equivalent to an individual cell). (B) Behaviour of a BisDC autoregulatory feedback loop on the distribution of BisDC protein levels per cell (histograms, n = 10,000 simulated cells) as a function of different binding (A1), unbinding (A2), production (A3) and degradation (A4) rate constants, starting from a uniformly distributed set of BisDC levels (input, in green). Black bar indicates the proportion of cells with *zero* output (i.e., non-activated circuit). Light blue: simulation example where BisDC levels go to zero and loop would die out. Dark blue: BisDC levels remain positive. (C) As for A, but for an architecture of BisR initiating *bisDC* expression, with different input distributions (uniformly low to high mean, or bimodal BisR input). Note how higher or bimodal BisR input is not expected to change the median BisDC quantity in active cells, but only the proportion of 'cells' with *positive* (magenta, bars) and *zero* state (black bars; n = 10,000 simulated cells; note different ordinate scales). (D) As for A, but for the complete cascade starting with TciR. Shown are regulatory factor level distributions from two different TciR starting distributions across 10,000 simulations; for BisR integrated between time points 10 and 20 (t10-20), and for BisDC after 100 time steps (t100). Bimodal expression of *zero* and *positive* states arises at the *bisR* node, but is further maintained to constant BisDC output as a result of the feedback loop. (E) Importance of the BisDC-feedback on the output of a downstream ('late') BisDC-dependent expressed protein, for a case of a stable and an unstable protein (n = 10,000 simulated cells). The online version of this article includes the following source data for figure 6:

Source data 1. *Figure 6* panels B–E: Model simulations and histogram data.

## BisDC-elements are widespread in other presumed ICEs

Pfam analysis detected a DUF2857-domain in the BisC protein, and further structural analysis using Phyre2 indicated significant similarities of BisC to FlhC (*Figure 1—figure supplement 2*). FlhC is a subunit of the master flagellar activator FlhDC of *E. coli* and *Salmonella* (*Claret and Hughes, 2000*; *Liu and Matsumura, 1994*). BisD carries a ParB domain, with a predicted DNA binding domain in the C-terminal portion of the protein (*Figure 1—figure supplement 2*). Although no FlhD domain was detected in BisD, in analogy to FhlDC we named the ICE*clc* activator complex BisDC, for bist-ability regulator subunits D and C.

Database searches showed that *bisDC* loci are also widespread among pathogenic and environmental *Gamma-* and *Beta-proteobacteria*, and are also found in some *Alphaproteobacteria* (*Figure 1—figure supplement 3*). Phylogenetic analysis using the more distantly related sequence from *Dickeya zeae* MS2 as an outgroup indicated several clear clades, encompassing notably *bisDC* homologs within genomes of *P. aeruginosa* and *Xanthomonas* (*Figure 1—figure supplement 4*). Several genomes contained more than one *bisDC* homolog, the most extreme case being *Bordetella petrii* DSM12804 with up to four homologs belonging to four different clades (*Figure 1—figure supplement 4*).

The gene synteny from *bisR* to *inrR* of ICE*clc* was maintained in several genomes (*Figure 1—figure supplement 3*), suggesting them being part of related ICEs with similar regulatory architecture. Notably, some of those are opportunistic pathogens, such *as P. aeruginosa*, *B. petrii*, *B. bronchiseptica*, or *X. citri*, and regions of high similarity to the ICE*clc* regulatory core extended to the well-known pathogenicity islands of the PAGI-2 (*Klockgether et al., 2007*) and PAGI-16 families (*Hong et al., 2016*; *Figure 1—figure supplement 3*). Several of the ICE*clc* core homologs carry genes suspected in virulence (e.g., filamentous hemagglutinin [*Sun et al., 2016*] encoded on the *P. aeruginosa* HS9 and Carb01-63 genomic islands), or implicated in acquired antibiotic resistance (e.g, multidrug efflux pump on the *A. xylosoxidans* NH44784-1996 element (*Miyazaki et al., 2015*), and carbapenem resistance on the PAGI-16 elements [*Hong et al., 2016*]). This indicates the efficacy of the ICE*clc* type regulatory control on the dissemination of this type of mobile elements, and consequently, on the distribution and selection of adaptive gene functions they carry.

## Discussion

ICEs operate a dual life style in their host, which controls their overall fitness as the integral of vertical descent (i.e., maintenance of the integrated state and replication with the host chromosome) and horizontal transfer (i.e., excision from the host cell, transfer and reintegration into a new host) (*Delavat et al., 2017*; *Delavat et al., 2016*; *Johnson and Grossman, 2015*). The decision for horizontal transfer is costly and potentially damages the host cell (*Delavat et al., 2016*; *Pradervand et al., 2014*), which is probably why its frequency of occurrence in most ICEs is fairly low (<10⁻⁵ per cell in a host cell population) (*Delavat et al., 2017*). Consequently, the mechanisms

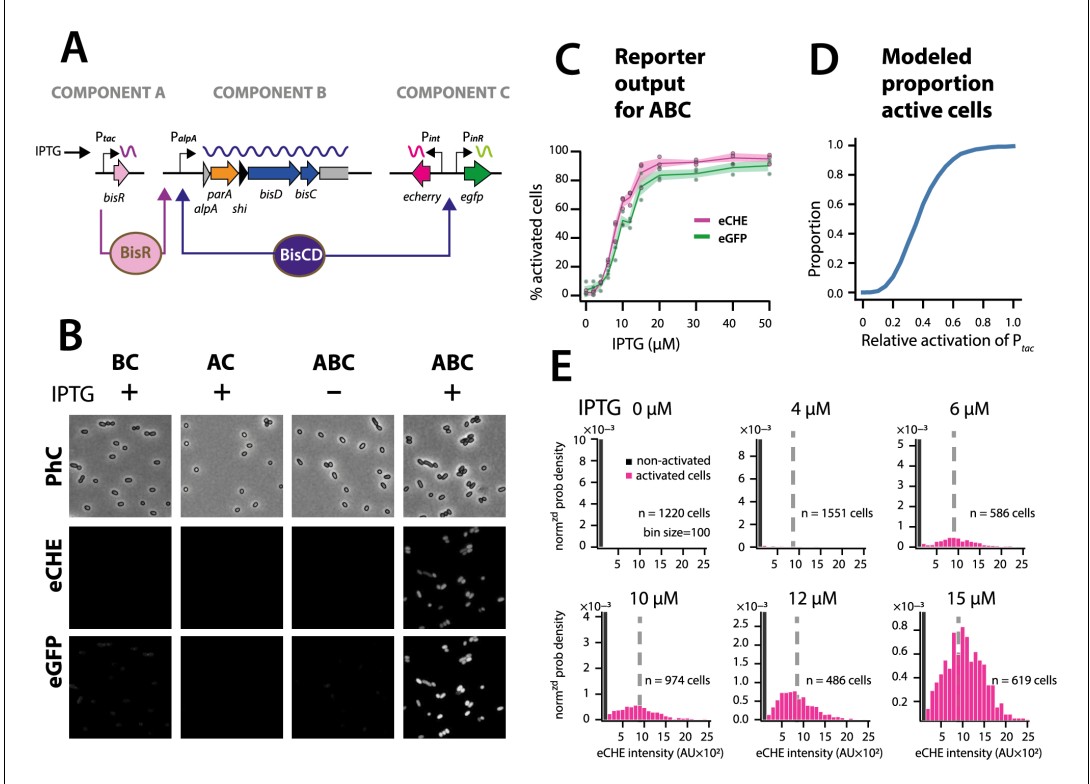

**Figure 7.** The ICE*clc* bistability generator is a scalable analog-digital converter. (**A**) Schematic representation of the three ICE*clc* components used to generate scalable bistable output in *P. putida* UWC1 without ICE. (**B**) Cell images of *P. putida* with the different bistability-generator components as indicated, induced in presence or absence of IPTG (0.1 mM). Fluorescence brightness scaled to between 300–1200. (**C**) Proportion of active cells (estimated from quantile-quantile plotting as in **Figure 2C**) as a function of IPTG concentration (same induction time for all). Lines correspond to the means from three biological replicates with transparent areas representing the standard deviation. (**D**) Modelled proportion of cells with *positive* output in the architecture of **Figure 6C** as a function of the relative BisR starting levels from $P_{tac}$. (**E**) Measured distributions (as normalized probability density) of eCherry fluorescence among the subpopulations of activated (magenta bars) and non-activated cells (black bars) at different IPTG concentrations, showing same subpopulation fluorescence median (dotted grey lines), as predicted in the stochastic model. AU, arbitrary units of fluorescence brightness at 500 ms exposure. *n* denotes the number of cells used to produce the histograms, summed from 6 to 12 images from a single replicate culture. Panels autoscaled to maximum ordinate.

The online version of this article includes the following source data and figure supplement(s) for figure 7:

**Source data 1.** *Figure 7* panel C: Mean fluorescence data.

**Source data 2.** *Figure 7* panel E: Single cell and histogram fluorescence data of IPTG induction.

**Figure supplement 1.** Scalable analog ($P_{tac}$) to digital bimodal ($P_{int}$) expression in *P. putida* with ICE*clc* bistability generator.

that initiate and ensure ICE horizontal transfer must have been selected to operate under extremely low opportunity with high success. In other words, they have been selected to maximize faithful maintenance of transfer competence development, once this process has been triggered in a host cell. One would thus expect such mechanisms to impinge on rare, perhaps stochastic cellular events, yielding robust output despite cellular gene expression and pathway noise. ICE*clc* is further particular in the sense that its transfer competence is initiated in cells during stationary phase conditions (*Miyazaki et al., 2012*), which restricts global transcription and activity, and may even profoundly alter the cytoplasmic state of the cell (*Parry et al., 2014*).

The results of our work here reveal that the basis for initiation and maintenance of ICE*clc* transfer competence in a minority of cells in a stationary phase population (*Reinhard et al., 2013*), originates in a multinode regulatory network that further includes a positive feedback loop. Genetic dissection, epistasis experiments and expression of individual components in *P. putida* devoid of the ICE showed that the network consists of a number of regulatory factors, composed of MfsR, TciR, BisR and BisDC, acting sequentially on singular (TciR, BisR) or multiple nodes (BisDC). The network has an 'upstream' branch controlling the initiation of transfer competence, a 'bistability generator' that

confines the input signal, and maintains the 'downstream' path of transfer competence to a dedicated subpopulation of cells (*Figure 1—figure supplement 1*).

The previously characterized *mfsR-marR-tciR* operon (*Pradervand et al., 2014*), whose transcription is controlled through autorepression by MfsR, is probably the main break on activation of the upstream branch. This was concluded from effects of deleting *mfsR*, which resulted in overexpression of TciR, and massively increased and deregulated ICE transfer even in exponentially growing cells (*Pradervand et al., 2014*). We showed here that TciR activates the transcription of a hitherto unrecognized transcription factor gene named *bisR*, but not of any further critical ICE*clc* promoters. Autorepression by MfsR in wild-type ICE*clc* results in low unimodal transcription from $P_{mfsR}$ (*Pradervand et al., 2014*) and therefore, likely, to low TciR levels in all cells. TciR appeared here as a weak activator of the *bisR* promoter, suggesting that only in a small proportion of cells it manages to trigger *bisR* transcription, as our model simulations further attested.

The BisR amino acid sequence revealed only very weak homology to known functional domains, thus making it the prototype of a new family of transcriptional regulators. In contrast to TciR, BisR was a very potent activator of its target, the *alpA* promoter. Model simulations suggested that BisR triggers and transmits the response in a scalable manner to the bistability generator, encoded by the genes downstream of $P_{alpA}$. Triggering of $P_{alpA}$ stimulated expression of (among others) two consecutive genes *bisD* and *bisC*, which code for subunits of an activator complex that weakly resembles the known regulator of flagellar synthesis FlhDC (*Claret and Hughes, 2000*; *Liu and Matsumura, 1994*). BisDC production was sufficient to activate the previously characterized bistable ICE*clc* promoters $P_{int}$ and $P_{inR}$, making it the key regulator for the 'downstream' branch (*Figure 1—figure supplement 1*). Importantly, BisDC was also part of a feedback mechanism activating transcription from $P_{alpA}$, and therefore, regulates its own production. Simulations and experimental data indicated that the feedback loop acts as a scalable analog-to-digital converter, transforming any *positive* input received from BisR into a dedicated cell that can regenerate sufficiently high BisDC levels to activate the complete downstream transfer competence pathway.

Bistable gene network architectures are characterized by the fact that expression variation is not resulting in a single mean phenotype, but can lead to two (or more) stable phenotypes - mostly resulting in individual cells displaying either one or the other phenotype (*Ferrell, 2012*; *Ferrell, 2002*; *Dubnau and Losick, 2006*). Importantly, such bistable states are an epigenetic result of the network functioning and do not involve modifications or mutations on the DNA (*Kussell and Leibler, 2005*; *Balázsi et al., 2011*). Bistable phenotypes may endure for a particular time in individual cells and their offspring, or erode over time as a result of cell division or other mechanism, after which the ground state of the network reappears. One can thus distinguish different steps in a bistable network: (i) the bistability *switch* that is at the origin of producing the different states, (ii) a *propagation* or *maintenance* mechanism and (iii) a *degradation* mechanism [11].

Some of the most well characterized bistable processes in bacteria include competence formation and sporulation in *Bacillus subtilis* (*Dubnau and Losick, 2006*). Differentiation of vegetative cells into spores only takes place when nutrients become scarce or environmental conditions deteriorate (*Veening et al., 2008*; *Veening et al., 2006*). Sporulation is controlled by a set of feedback loops and protein phosphorylations, which culminate in levels of the key regulator SpoOA ~P being high enough to activate the sporulation genes (*Dubnau and Losick, 2006*). In contrast, bistable competence formation in *B. subtilis* is generated by feedback transcription control from the major competence regulator ComK. Stochastic variations among ComK levels in individual cells, ComK degradation and inhibition by ComS, and noise at the *comK* promoter determine the onset of *comK* transcription, which then reinforces itself because of the feedback mechanism (*Süel et al., 2006*; *Maamar et al., 2007*). Initiation and maintenance of the ICE*clc* transfer competence pathway thus resembles DNA transformation competence in *B. subtilis* in its architecture of an auto-feedback loop (BisDC vs ComK). However, the switches leading to bistability are different, with ICE*clc* depending on a hierarchy of transcription factors (MfsR, TciR and BisR), and transformation competence being a balance of ComK degradation and inhibition of such degradation (*Süel et al., 2006*; *Maamar et al., 2007*). ICE*clc* bistability architecture is clearly different from the well-known double negative feedback control exerted by, for example the phage lambda lysogeny/lytic phase decision in *E. coli* (*Arkin et al., 1998*; *Bednarz et al., 2014*). That switch entails essentially a balance of the counteracting transcription factors CI, CII and Cro (*Arkin et al., 1998*; *Bednarz et al., 2014*). Interestingly, other ICEs of the SXT/R391 family carry this typical double negative feedback loop architecture,

which may therefore control their (bistable) activation (*Poulin-Laprade and Burrus, 2015*; *Bellanger et al., 2008*; *Beaber and Waldor, 2004*; *Poulin-Laprade et al., 2015*). Given the low frequencies of conjugative transfer of many different elements (*Delavat et al., 2017*), bistability activation mechanisms may be much more widespread than assumed.

Mathematically speaking, the ICE*clc* transfer competence regulatory architecture has two states, one of which is *zero* (inactive) and the other with a *positive* value (activation of transfer competence). Stochastic modelling suggested that the feedback loop maintains *positive* output during a longer time period than in its absence (although it will drop to *zero* at infinite time). Previous experimental data suggested that the tc cells indeed do not return to a silent ICE*clc* state, but become irreparably damaged, arrest their division (*Takano et al., 2019*) and wither (*Reinhard et al., 2013*). However, because their number is proportionally low, there is no fitness cost on the population carrying the ICE (*Delavat et al., 2016*; *Gaillard et al., 2008*). The advantage of prolonged feedback output seems that constant levels of the BisCD regulator can be maintained, allowing coordinated and organized production of the components necessary for the ICE*clc* transfer itself. This would consist of, for example, the relaxosome complex responsible for DNA processing at the origin(s) of transfer, and the mating pore formation complex (*Carraro and Burrus, 2015*). Because *Pseudomonas* cells activate ICE*clc* transfer competence upon entry in stationary phase, the feedback loop may have a critical role to ensure faithful completion of the transfer competence pathway during this period of limiting nutrients, and to allow the ICE to excise and transfer from tc cells once new nutrients become available (*Delavat et al., 2016*).

Although our results were conclusive on the roles of the key regulatory factors (MfsR, TciR, BisR, BisDC), there may be further auxiliary and modulary factors, and environmental cues that influence the transfer competence network. For example, we previously found that deletions in the gene *inrR* drastically decreased ICE*clc* transfer capability by 45–fold and reduced reporter gene expression from $P_{int}$ (*Minoia et al., 2008*). Expression of InrR alone, however, did not show any direct activation of $P_{int}$, $P_{inR}$ or $P_{alpA}$, and InrR is thus unlikely to be a direct transcription activator protein. Our results also indicated that induction of AlpA may repress output from the $P_{alpA}$ promoter, and modulate the feedback loop that is initiated by BisR and maintained by BisDC. Furthermore, although induction of *bisDC* was sufficient to activate expression from $P_{alpA}$, it was enhanced through an as yet uncharacterized mechanism involving its upstream regions. Previous results also highlighted the implication of the stationary phase sigma factor RpoS for $P_{inR}$ activation (*Figure 1B*; *Miyazaki et al., 2012*), which may be more generally important for other ICE*clc* regulatory promoters as well. Unraveling these details in future work will be important for a full understanding of the generation and maintenance of bistability of the ICE*clc* family of elements, and its role in effective horizontal dissemination.

Phylogenetic analyses showed the different ICE*clc* regulatory loci (i.e., *bisR-alpA-bisDC-inrR*) to be widely conserved in Beta- and Gammaproteobacteria, with only few small variations in regulatory gene configurations. Most likely, these regions are part of ICE*clc*-like elements in these organisms, several of which have been detected previously (*Miyazaki et al., 2015*; *Miyazaki, 2011a*; *Gaillard et al., 2006*). They are further part of PAGI-2 (*Klockgether et al., 2007*) and PAGI-16 family genomic islands in *P. aeruginosa* clinical isolates (*Hong et al., 2016*) that have been implicated in the distribution of virulence factors and antibiotic resistance elements. The ICE*clc* regulatory cascade for transfer competence thus seems widely conserved, controlling horizontal dissemination of this important class of bacterial conjugative elements.

## Materials and methods

### Key resources table

| Reagent type (species) or resource | Designation | Source or reference | Identifiers | Additional information |
|---|---|---|---|---|
| Gene (*Pseudomonas knackmussii* ICE*clc*) | ICE*clc* | PMID:16484212 | GenBank: AJ617740.2 | Full ICEclc sequence |

*Continued on next page*

*Continued*

| Reagent type (species) or resource | Designation | Source or reference | Identifiers | Additional information |
|---|---|---|---|---|
| Gene (*Pseudomonas knackmussii* ICE*clc*) | *tciR* | PMID:24945944 | GenBank: CAE92867.2 | Transcriptional regulator of ICE*clc* |
| Gene (*Pseudomonas knackmussii* ICE*clc*) | *bisR; orf101284* | this paper | GenBank: CAE92957.1 | Transcriptional regulator of ICE*clc* |
| Gene (*Pseudomonas knackmussii* ICE*clc*) | *bisD; parB; orf98147* | this paper | GenBank: CAE92953.1 | Subunit D of the transcriptional regulator complex BisCD of ICE*clc* |
| Gene (*Pseudomonas knackmussii* ICE*clc*) | *bisC; orf97571* | this paper | GenBank: CAE92952.1 | Subunit C of the transcriptional regulator complex BisCD of ICE*clc* |
| Gene (plasmid pZS2FUNR) | *echerry* | PMID:19098098 | | Red fluorescent protein gene used for the miniTn7:*Ptac-echerry* reporter |
| Strain, strain background (*Pseudomonas putida*) | UWC1 | PMID:2604401 | NCBI: txid1407054 | Background strain for ICE*clc* transfer and mutagenesis experiments |
| Strain, strain background (*Pseudomonas putida*) | 2737 | PMID:21255116 | | UWC1 carrying ICE*clc* in tRNAGly |
| Strain, strain background (*Pseudomonas putida*) | UWCGC | PMID:21255116 | | Recipient strain used for conjugation transfer experiments |
| Strain, strain background (*Escherichia coli*) | DH5αλpir | PMID:10610816 | | Cloning strain |
| Recombinant DNA reagent | pME6032; pME (plasmid) | PMID:11807065 | GenBank: DQ645594.1 | Broad-host range cloning vector, lacIq-Ptac expression |
| Recombinant DNA reagent | miniTn5 (plasmid; transposon) | PMID:21342504 (RRID:Addgene_60487) | GenBank: HQ908071.1 | Transposon suicide vector for gene reporter constructs |
| Recombinant DNA reagent | miniTn7 (plasmid; transposon) | PMID:15908923 (RRID:Addgene_63121) | GenBank: AY619004.1 | Transposon suicide vector for single copy insertions |
| Recombinant DNA reagent | miniTn7:*Ptac* (plasmid; transposon) | PMID:15908923 | GenBank: AY599234.1 | Transposon vector used for miniTn7:*Ptac-echerry* reporter |
| Recombinant DNA reagent | miniTn5:*PinR-egfp/Pint-echerry* | PMID:19098098 | | Dual single copy insertion reporter system for ICE*clc* bistable activity |
| Sequence-based reagent | DNA fragment containing $P_{alpA}$, *alpA*, *parA*, *shi* and the 5' part of *bisD* | ThermoFisher Scientific | | Synthetic DNA fragment |
| Commercial assay or kit | Nucleospin Plasmid Kit | Macherey-Nagel | Macherey-Nagel: 740588.50 | Plasmid purification |
| Commercial assay or kit | Nucleospin Gel and PCR Clean-up Kit | Macherey-Nagel | Macherey-Nagel: 740609.50 | PCR fragment purification |
| Commercial assay or kit | Quick-Fusion Cloning Kit | Bimake | Bimake: B22611 | Generation of recombinant vectors |
| Chemical compound, drug | IPTG; isopropyl β-D-1-thiogalactopyranoside | Sigma-Aldrich | Sigma-Aldrich: I5502 | Used for induction of $P_{tac}$ promoter |

*Continued on next page*

*Continued*

| Reagent type (species) or resource | Designation | Source or reference | Identifiers | Additional information |
|---|---|---|---|---|
| Chemical compound, drug | 3-CBA; 3-chlorobenzoate (3-chlorobenzoic acid) | Sigma-Aldrich | Sigma-Aldrich: C24604 | Specific carbon source for selection of ICE*clc* in *Pseudomonas* |
| Chemical compound, drug | Sodium succinate; succinate (Sodium succinate dibasic hexahydrate) | Sigma-Aldrich | Sigma-Aldrich: 14170 | General carbon source for growth of *Pseudomonas* |
| Software, algorithm | MEGA7 | PMID:27004904 | | Phylogenetic analysis |
| Software, algorithm | Visiview | Visitron systems GMbH | https://www.visitron.de /products/visiviewr-software.html | Microscopy images acquisition |
| Software, algorithm | ImageJ | PMID:22930834 | | Image processing |
| Software, algorithm | MatLab (v 2016a) | Mathworks | | Data treatment |
| Software, algorithm | R ggplot | Hadley Wickham | https://ggplot2 .tidyverse.org/ | Data visualization |
| Software, algorithm | Julia; Differential Equations.jl package | DOI:http://doi.org/ 10.5334/jors.151 | | Mathematical model |
| Other | Minimum media; MM (culture media) | PMCID:PMC494262 | *Supplementary file 2* | Carbon source-free base for minimal media |
| Other | Bio-Rad Gene Pulser Xcell | Biorad | Biorad: 165–2660 | Device used for electro-transformation of bacterial strains |
| Other | Zeiss Axioplan II microscope (Carl Zeiss); EC 'Plan-Neofluar' 100x/1.3 Oil Pol Ph3 M27 objective lens (Carl Zeiss); SOLA SE light engine (Lumencor); SPOT Xplorer slow-can charge coupled device camera 1.4 Megapixels monochrome w/o IR (Diagnostic Instruments) | Carl Zeiss; Lumencor; Diagnostic instruments | Carl Zeiss: Axioplan II; Carl Zeiss: 420491-9910-000; Lumencor: SOLA 6-LCR-SC; Diagnostic instrument: XP2400 | Microscope for single cell phase contrast and epifluorescence imaging |
| Other | 0.2–μm cellulose acetate filters | Sartorius | Sartorius: 11107–25 N | filters for conjugative transfer |

## Strains and growth conditions

Bacterial strains and plasmid constructions used in this study are shortly described in *Table 1* and with more detail in *Supplementary file 1*. Strains were routinely grown in Luria broth (10 g l$^{-1}$ Tryptone, 10 g l$^{-1}$ NaCl and 5 g l$^{-1}$ Yeast extract, LB Miller, Sigma Aldrich) at 30°C for *P. putida* and 37°C for *E. coli* in an orbital shaker incubator, and were preserved at –80°C in LB broth containing 15% (*v/ v*) glycerol. Reporter assays and transfer experiments were carried out with cells grown in type 21C minimal media (MM, *Supplementary file 2*; *Gerhardt, 1981*) supplemented with 10 mM sodium succinate or 5 mM 3-chlorobenzoate (3-CBA). Antibiotics were used at the following concentrations: ampicillin (Ap), 100 μg ml$^{-1}$ for *E. coli* and 500 μg ml$^{-1}$ for *P. putida*; gentamicin (Gm), 10 μg ml$^{-1}$ for *E. coli*, 20 μg ml$^{-1}$ for *P. putida*; kanamycin (Kn), 50 μg ml$^{-1}$; tetracycline (Tc), 12 μg ml$^{-1}$ for *E. coli*, 100 μg ml$^{-1}$ or 12.5 μg ml$^{-1}$ for *P. putida* grown in LB or MM, respectively. Genes were induced from $P_{tac}$ by supplementing cultures with 0.05 mM isopropyl β-D-1-thiogalactopyranoside (IPTG; or else at the indicated concentrations).

## Molecular biology methods

Plasmid DNA was purified using the Nucleospin Plasmid kit (Macherey-Nagel) according to manufacturer's instructions. All enzymes used in this study were purchased from New England Biolabs. PCR

**Table 1.** Strains and plasmids used in this study.

| Strains or plasmids | Relevant genotype or phenotype | References |
|---|---|---|
| *E. coli* | | |
| DH5αλpir | *endA*1 *hsdR*17 *glnV*44 (=*supE*44) *thi*-1 *recA*1 *gyrA*96 *relA*1 φ80d*lac* Δ(*lacZ*)M15 Δ(*lacZYA-argF*) U169 *zdg*-232::Tn10 *uidA*::*pir*+ | *Platt et al., 2000* |
| *P. putida* UWC1 | plasmid-free derivative of *P. putida* KT2440 (Rif) | *McClure et al., 1989* |
| UWCGC | Single copy integration of *lacI*-less $P_{tac}$ promoter controlling *echerry* expression (Gm) | *Miyazaki and van der Meer, 2011b* |
| ICE*clc* | ICE*clc* copy integrated into tRNA$^{gly-5}$ (3-CBA) | *Miyazaki and van der Meer, 2011b* |
| ICE*clc* Δ*tciR* | *tciR* (*orf17162*) derivative mutant of ICE*clc* (3-CBA) | *Pradervand et al., 2014* |
| ICE*clc* Δ*bisR* | *bisR* (*orf101284*) derivative mutant of ICE*clc* (3-CBA) | This work |
| ICE*clc* Δ*bisD* | *bisD* (orf98147) derivative mutant of ICE*clc* (3-CBA) | This work |
| miniTn7::$P_{inR}$-*egfp* | Single copy chromosomal integration of $P_{inR}$ promoter fused to *egfp* (Gm) | This work |
| miniTn7::$P_{alpA}$-*egfp* | Single copy chromosomal integration of $P_{alpA}$ promoter fused to *egfp* (Gm) | This work |
| miniTn5:: $P_{bisR}$-*egfp* | Single copy chromosomal integration of $P_{bisR}$ promoter fused to *egfp* (Kn) | This work |
| miniTn5:: $P_{int}$-*echerry*/$P_{inR}$ egfp (C) | Single copy chromosomal integration of a dual reporter $P_{int}$ and $P_{inR}$ promoter fused to *echerry* and *egfp*, respectively (Kn) | *Minoia et al., 2008* |
| miniTn7::$P_{tac}$-*bisR* (A) | Single copy chromosomal integration of $P_{tac}$ promoter fused to *bisR* (Gm) | This work |
| miniTn7::$P_{tac}$-*echerry* | Single copy chromosomal integration of $P_{tac}$ promoter fused to *echerry* (Gm) | This work |
| Plasmids | | |
| pME6032 | pVS1-p15A shuttle vector carrying the *lacI*$^q$-$P_{tac}$ expression system (Tc) | *Heeb et al., 2000* |
| pME*tciR* | pME6032 derivative allowing IPTG-controlled expression of *tciR* (Tc) | This work |
| pME*bisR* | pME6032 derivative allowing IPTG-controlled expression of *bisR* (Tc) | This work |
| pME*bisC* | pME6032 derivative allowing IPTG-controlled expression of *bisC* (Tc) | This work |
| pME*bisD* | pME6032 derivative allowing IPTG-controlled expression of *bisD* (Tc) | *Reinhard et al., 2013* |
| pME*bisDC* | pME6032 derivative allowing IPTG-controlled expression of *bisDC* (Tc) | This work |
| pME*parA* | pME6032 derivative allowing IPTG-controlled expression of *parA* (Tc) | *Reinhard et al., 2013* |
| pME*parA-shi-bisD* | pME6032 derivative allowing IPTG-controlled expression of *parA*, *shi* and *bisD* (Tc) | *Reinhard et al., 2013* |

*Table 1 continued on next page*

*Table 1 continued*

| Strains or plasmids | Relevant genotype or phenotype | References |
|---|---|---|
| pME*bisC96* | pME6032 derivative allowing IPTG-controlled expression of *bisC* and *96323* (Tc) | This work |
| pME*alpA* | pME6032 derivative allowing IPTG-controlled expression of *alpA* (Tc) | This work |
| pME*inrR* | pME6032 derivative allowing IPTG-controlled expression of *inrR* (Tc) | This work |
| pME*reg* | pME6032 derivative allowing IPTG-controlled expression of the *alpA-inrR* loci (Tc) | This work |
| pME*reg*ΔalpA | pME6032 derivative allowing IPTG-controlled expression of the *parA-inrR* loci (Tc) | This work |
| pME*reg*ΔalpAΔP | pME*reg*ΔalpA derivative lacking 3' half of *96323*, *95213* and *inrR* (Tc) | This work |
| pME*reg*ΔalpAΔA | pME*reg*ΔalpA derivative lacking the *bisC*, *96323*, *95213* and *inrR* (Tc) | This work |
| pME*reg*ΔP | pME*reg* derivative lacking 3' half of *96323*, *95213* and *inrR* (Tc) | This work |
| pME*reg*ΔA | pME*reg* derivative lacking the *bisC*, *96323*, *95213* and *inrR* (Tc) | This work |
| pME*bg* | *lacI*^q-*P*_{tac}-less pME6032 derivative carrying the *P*_{alpA}-*inrR* loci (Tc) | This work |
| pME*bg_short* (comp B) | pME*bg* derivative lacking *96323*, *95213* and *inrR* (Tc) | This work |
| pUX-BF13 | helper plasmid for integration of Tn7 (Ap) | *Heeb et al., 2000* |

3-chlorobenzoate (3-CBA); Ampicillin (Ap); gentamycin (Gm); kanamycin (Kn); rifampicin (Rf); tetracycline (Tc).

(A), (B) and (C) refer to components of the reconstituted bistability generator.

For strain numbers, see **Supplementary file 1**.

reactions were carried out with primers described in *Supplementary file 3*. PCR products were purified using Nucleospin Gel and PCR Clean-up kits (Macherey-Nagel) according to manufacturer's instructions. *E. coli* and *P. putida* were transformed by electroporation as described by *Dower et al., 1988*. in a Bio-Rad GenePulser Xcell apparatus set at 25 μF, 200 V and 2.5 kV for *E. coli* and 2.2 kV for *P. putida* using 2 mm gap electroporation cuvettes (Cellprojects). All constructs were verified by DNA sequencing (Eurofins).

## Cloning of regulatory pathway elements

Different ICE*clc* gene configurations were cloned in *P. putida* with or without ICE*clc*, and further with different promoter-reporter fusions, using the broad host-range vector pME6032, allowing IPTG-controlled expression from the LacI^q-*P*_{tac} promoter (*Koch et al., 2001*; *Table 1*). Genes *tciR*, *bisR*, *bisC*, *bisDC*, *bisC+96323*, *alpA* and *inrR* were amplified using primer pairs as specified in *Supplementary file 3*, with genomic DNA of *P. putida* UWC1-ICE*clc* as template. Amplicons were digested by EcoRI and cloned into EcoRI-digested pME6032 using T4 DNA ligase, producing after transformation the plasmids listed in *Table 1*. The 6.4 kb ICE*clc* left-end fragment encompassing *parA-inrR* was recovered from pTCB177 (*Sentchilo et al., 2003*) and cloned into pME6032 (producing pME*reg*ΔalpA, *Supplementary file 1*). An *alpA-parA-shi-bisD'* fragment was amplified by PCR (*Supplementary file 2*) and cloned into pME6032 using EcoRI restriction sites (*Supplementary file 1*). The resulting plasmid was digested with SalI and the 4.8 kb fragment containing the *P*_{tac} promoter, *alpA-parA-shi-bisD'* was recovered and used to replace the *parA-shi-parB* part of pME*reg*ΔalpA. This generated a cloned fragment encompassing *alpA* all the way to *inrR* (pME*reg*, *Supplementary file 1*). Further 3' deletions removing *orf96323-inrR* or *bisC-inrR* were generated by

PstI and AfeI digestion and religation (*Supplementary file 1*). A DNA fragment containing *P*$_{alpA}$, *alpA*, *parA*, *shi* and the 5' part of *bisD* was synthesized (ThermoFisher Scientific), and ligated by Quick-Fusion cloning (Bimake) into pMEregΔalpA digested with PmlI and BamHI to remove the part containing *lacI*$^q$, *P*$_{tac}$, *parA*, *shi* and *bisD*. This plasmid was then digested by PstI to remove *orf96323-inrR* and religated (*Supplementary file 1*).

Deletions of *bisR* or *bisD* in ICE*clc* were constructed using the two-step seamless chromosomal gene inactivation technique as described elsewhere (*Martínez-García and de Lorenzo, 2011*).

## Reporter gene constructs

Activation of key ICE*clc* promoters was determined in strains with a single-copy chromosomal insertions to promoterless *egfp* or *echerry* genes, for most cases delivered by a suicide miniTn*7* system at a fixed unique position (*Table 1*). In other cases, particularly in combination with other single-copy inserted gene fragments, we used miniTn*5* random delivery. The promoter regions upstream of *bisR* or *alpA* were amplified in the PCR (*Supplementary file 3*) and cloned into the promoterless *egfp* reporter miniTn*5* delivery plasmid pBAM1 (*Martínez-García et al., 2011*) or into a pUC18-derived miniTn*7* delivery plasmid (*Choi et al., 2005*). The *P*$_{inR}$-*egfp* insert was recovered from the miniTn*5*-based reporter system (*Minoia et al., 2008*) using HindIII and KpnI, and subsequently cloned into pUC18miniTn*7* digested by the same enzymes. The dual miniTn*5*::*P*$_{inR}$-*egfp*/*P*$_{int}$-*echerry* reporter has been described previously (*Minoia et al., 2008*). A miniTn*7*::*P*$_{tac}$-*echerry* reporter was reconstructed from pZS2FUNR (*Minoia et al., 2008*) and the general miniTn*7*:*P*$_{tac}$ suicide delivery vector (*Choi et al., 2005*; *Supplementary file 2*). All reporter constructs were integrated in single copy into the chromosomal *attB*$_{Tn7}$ site of *P. putida* by using pUX-BF13 for miniTn*7*, or randomly for miniTn*5*-based constructs (*Martínez-García et al., 2011*; *Koch et al., 2001*), in which case three independent clones were recovered, stored and analysed. The intactness of the inserted reporter constructs was verified by PCR amplification and sequencing.

## ICE*clc* transfer assays

ICE*clc* transfer was tested with 24-h-succinate-grown donor and recipient cultures. Cells were harvested by centrifugation of 1 ml (donor) and 2 ml culture (recipient, Gm-resistant *P. putida* UWCGC) for 3 min at 1200 × *g*, washed in 1 ml of MM without carbon substrate, centrifuged again and finally resuspended in 20 µl of MM. Donor or recipient alone, and a donor-recipient mixture were deposited on 0.2–µm cellulose acetate filters (Sartorius) placed on MM succinate agar plates, and incubated at 30°C for 48 hr. The cells were recovered from the filters in 1 ml of MM and serially diluted before plating. Donors, recipients and exconjugants were selected on MM agar plates containing appropriate antibiotics and/or carbon source (3-CBA). Transfer frequencies are reported as the mean of the exconjugant colony forming units compared to that of the donor in the same assay.

## Molecular phylogenetic analysis

BisDC phylogeny was inferred from 148 aligned homolog amino acid sequences by using the Maximum Likelihood method based on the Tamura-Nei model (*Tamura and Nei, 1993*), eliminating positions with less than 95% site coverage. The final dataset was aligned using MEGA7 (*Kumar et al., 2016*) and contained a total of 2091 positions. Initial tree(s) for the heuristic search were obtained automatically by applying Neighbour-Joining and BioNJ algorithms to a matrix of pairwise distances estimated using the Maximum Composite Likelihood (MCL) approach, and then selecting the topology with superior log likelihood value.

## Fluorescent reporter assays

For quantification of eGFP and eCherry fluorescence in single cells, *P. putida* strains were cultured overnight at 30°C in LB medium. The overnight culture was diluted 200 fold in 8 ml of MM supplemented with succinate (10 mM) and appropriate antibiotic(s), and grown at 30°C and 180 rpm to stationary phase. 150 µl of culture were then sampled, vortexed for 30 s at max speed, after which drops of 5 µl were deposited on a regular microscope glass slide (VWR) coated with a thin film of 1% agarose in MM. Cells were covered with a 24 × 50 mm cover slip (Menzel-Gläser) and imaged immediately with a Zeiss Axioplan II microscope equipped with an EC Plan-Neofluar 100×/1.3 oil objective lens (Carl Zeiss), and a SOLA SE light engine (Lumencor). A SPOT Xplorer slow-can charge

coupled device camera (1.4 Megapixels monochrome w/o IR; Diagnostic Instruments) fixed on the microscope was used to capture images. Up to ten images at different positions were acquired using Visiview software (Visitron systems GMbH), with exposures set to 40 ms (phase contrast, PhC) and 500 ms (eGFP and eCherry). Cells were automatically segmented on image sets using procedures described previously (*Delavat et al., 2016*), from which their fluorescence (eGFP or eCherry) was quantified. Subpopulations of tc cells were quantified using quantile-quantile-plotting as described previously (*Reinhard and van der Meer, 2013*). Fluorescent images for display were scaled to the same brightness in ImageJ (*Schneider et al., 2012*) as indicated, saved as 8-bit gray tiff-files and cropped to the display area in Adobe Photoshop (Adobe, 2020).

## Statistical analysis

Fluorescent reporter intensities were compared among biological triplicates. In case of mini-Tn5 insertions, this involved three clones with potentially different insertion sites, each measured individually. For mini-Tn7 inserted reporter constructs, we measured three biological replicates of a unique clone. Expression differences between mutants and a strain with the same genetic background but carrying the empty pME6032 plasmid were tested on triplicate means of individual median or 75th percentile values in a one-sided t-test (the hypothesis being that the mutant expression is higher than the control). Comparison of 75th percentiles rather than median or mean is justified when populations are extremely skewed, as we previously demonstrated (*Reinhard and van der Meer, 2013*). Coherent simultaneous data series were tested for significance of reporter expression or transfer frequency differences in ANOVA, followed by a post-hoc Tukey test. Quantile-quantile plots were produced in MatLab (v 2016a), violin -boxplots by using *ggplot2* in R.

## Mathematical model of ICE*clc* activation

ICE*clc* activation was simulated as a series of stochastic events in different network configurations (as schematically depicted in *Figure 6A*, *Supplementary file 4*). TciR, BisR, BisDC and protein output levels were then simulated using the Gillespie algorithm (*Gillespie, 1977*; *Gillespie, 1976*), implemented in *Julia* using its DifferentialEquations.jl package (*Rackauckas and Nie, 2017*). 10,000 individual simulations (each simulation corresponding to a single 'cell') were conducted per network configuration during 100 time steps, during or after which the remaining protein levels were counted and summarized. The code for the mathematical implementation is provided in the *Source code 1*.

# Acknowledgements

The authors thank Fabrice Merz and Noëmie Matthey for their help in technical parts of this study. We thank Aleksandar Vjestica, Roxane Moritz and Andrea Daveri for critical reading. The work was supported by Swiss National Science Foundation grant to JvdM 31003A_175638 and by a SystemsX. ch Interdisciplinary grant to CM and JvdM.

The funders had no role in study design, data collection and analysis, decision to publish, or preparation of the manuscript.

# Additional information

### Funding

| Funder | Grant reference number | Author |
| --- | --- | --- |
| Swiss National Science Foundation | 31003A_175638 | Jan Roelof van der Meer |
| SystemsX.ch | Interdisciplinary Grant | Christian Mazza |

The funders had no role in study design, data collection and interpretation, or the decision to submit the work for publication.

## Author contributions

Nicolas Carraro, Conceptualization, Data curation, Formal analysis, Validation, Investigation, Methodology, Writing - original draft, Writing - review and editing; Xavier Richard, Software, Validation, Visualization, Methodology, Writing - review and editing; Sandra Sulser, Conceptualization, Investigation, Methodology, Writing - review and editing; François Delavat, Data curation, Investigation, Methodology, Writing - review and editing; Christian Mazza, Conceptualization, Software, Formal analysis, Supervision, Funding acquisition, Writing - review and editing; Jan Roelof van der Meer, Conceptualization, Data curation, Formal analysis, Supervision, Funding acquisition, Validation, Investigation, Visualization, Methodology, Writing - original draft, Project administration, Writing - review and editing

## Author ORCIDs

Nicolas Carraro http://orcid.org/0000-0001-6364-547X
François Delavat https://orcid.org/0000-0002-5985-4583
Jan Roelof van der Meer https://orcid.org/0000-0003-1485-3082

## Decision letter and Author response

Decision letter https://doi.org/10.7554/eLife.57915.sa1
Author response https://doi.org/10.7554/eLife.57915.sa2

## Additional files

### Supplementary files

- Source code 1. Julia code for the stochastic modelling of bistability.
- Supplementary file 1. Full strain list.
- Supplementary file 2. Composition of minimal growth medium.
- Supplementary file 3. List of Primers.
- Supplementary file 4. Stochastic model for bistability architectures.
- Transparent reporting form

## Data availability

All data generated or analysed during this study are included in the manuscript and supporting files. Source data files have been provided.

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
