## [Decision Letter]

**Acceptance summary:**

Integrative Conjugative elements (ICEs) play an important role in bacterial adaptation as they can horizontally transfer phenotypic traits between bacteria, ranging from biodegradation of aromatic compounds to antibiotic resistance and virulence. Their low rate of excision from the chromosome and transfer to other cells has been attributed to the existence of two mutually exclusive stable states within the population: the transfer-competent and non-active state. For a particular family of these ICEs, ICE*clc*, the regulatory basis for the activation of this so-called bistable transfer competence pathway has remained largely elusive. This paper elegantly combines various genetic tools and stochastic modeling to identify these regulatory mechanisms, discovered a transcription factor that is part of a new regulator family, and showed that the feedback loop they described acts as a converter of a unimodal input to a bistable output. It will be interesting to learn from future studies how important biological bistability is in the horizontal transfer of genes via conjugation mediated by various plasmids and ICEs.

**Decision letter after peer review:**

Thank you for submitting your article "An analog to digital converter controls bistable transfer competence of a widespread integrative and conjugative element" for consideration by *eLife*. Your article has been reviewed by three peer reviewers, including Eva Top as the Guest Editor and Reviewer #1, and the evaluation has been overseen by Naama Barkai as the Senior Editor. The following individual involved in review of your submission has agreed to reveal their identity: Rafael Silva-Rocha (Reviewer #3).

The reviewers have discussed the reviews with one another and the Guest Editor has drafted this decision to help you prepare a revised submission.

Summary:

This is a very thorough and elegant study characterizing the different components of the regulatory cascade governing the bistable switch between non-active and transfer competent cells of *Pseudomonasputida* carrying the Integrative and Conjugative Element ICE*clc*. The authors used a number of promoter reporter constructions together with mathematical modeling to investigate the key regulatory components of the ICE*clc* conjugative element in *P. putida*. They developed an original conceptual mathematical model to test several hypotheses and then tested them experimentally. They demonstrated that the feedback loop regulating the third node of the cascade maintains production of the regulator during a longer period, which enables the activation of the transfer competence pathway (referred as an "analog to digital converter"). Such regulation is likely widespread in other gamma- and beta-proteobacteria.

Essential revisions:

1) Along with the description of the underlined mechanisms controlling this process, a novel transcription factor (BisR) was described. Perhaps the new regulator could be emphasized more in the Abstract. The work is very complete and well-performed. The authors do a nice job of walking the reader through the various genetic manipulations that were needed to draw the conclusions on a complex regulatory system, especially in the first part of the Results.

2) The manuscript could be improved as a publication for *eLife* if the authors argue more than they do now about the general interest of their work, the possible importance of these elements in pathogens, and respond to the few methodological issues raised. A few particular suggestions are made below.

3) Since the *clc* element transfers by conjugation much like conjugative plasmids, and several plasmids, like those of the IncF family, also seem to transfer only from a limited number of cells (or at least the rates of transfer are very low), is there any information on whether or not conjugation mediated by some groups of plasmids may also be controlled by such a complex system? Given the broad readership of *eLife*, it would be helpful to broaden the discussion to horizontal (or at least conjugative) transfer of other genetic elements.

4) Though the role of RpoS is intriguing, and suggests some kind of stress response. Can more be said about that in this study, even though it was not the focus but seems to be critical?

5) Subsection “A new regulator BisDC is the last step in the activation cascade”: I did not quite understand at first why complementation of a bisD deletion mutant with bisCD did not result in similar frequencies of transfer as in the wild-type or other deletion mutants. It would be helpful to elaborate a bit more on this concept of 'reinforcement' at this stage in the paper, more specifically how these findings led to that conclusion. Later on, when the positive feedback is demonstrated, it would be helpful to go back and explain this result.

6) The only place where I was a bit lost during the first read-through was the evidence of a bistable output (see below). I think the average reader will be puzzled by the fact that you equate bistable with 'digital' and a single signal with 'analog'. Some explanation is warranted here. Moreover, is this the only circuit working as analog-to-digital converter in bacteria? More comparison to other systems in bacteria was missing. Can you elaborate features that seem unique so far in ICE*clc* and those that are similar to other systems.

7) In general terms, the authors did not put much emphasis on the role of AlpA on the modulation of the circuit. What is this element? Which could be the potential mechanisms for its effect on PalpA? It is clear this element is relevant for the systems, but this has been only superficially mentioned in the work. It would strengthen the work if this gene would be investigated a bit more, at least by bioinformatics.

8) The authors exploit the phylogenetic distribution of ICE*clc* in several organisms. Have this element (or similar) been described associated with pathogenesis or antibiotic resistance? It was mentioned early at the Introduction the association of ICE elements with antibiotic elements, but has the key regulatory elements of ICE*clc* been identified in pathogenic bacteria or associated with virulence? This information would enhance the general interest of the work beyond environmental bacteria and biodegradation.

9) Why do the authors use 75th of relative fluorescence? What is the rational for that? What would happen if relative fluorescence for all cells were used?

10) The authors used different single-copy reporter systems to investigate promoter activity. For miniTn5, 3 independent clones were used. Yet, even in this case the system will not be isogenic. It would be better to have all reporter systems using the same insertion locus, such as miniTn7, to have a faithful composition between strains. In terms of methodology, for me it’s is the only concern that I have.

11) One concern is that the authors used two approaches to decipher the regulatory cascade of this ICE. The first one relied on the mutagenesis of putative regulator genes of ICE*clc* and complementation by genes cloned in plasmids. The second one consisted in the ectopic expression of individual and combinations of suspected regulatory elements in a host without ICE*clc* and study of the expression of single chromosomal copy of transcriptional fusions. Both methods rely on ectopic expression of the regulator genes and thus in an "off-ground" analysis, i.e. not in the in situ context of transcription of ICE genes. Thus potential regulation elements can be missed: secondary structures of long RNA transcripts, competition between regulators, dosage of regulators versus promoters etc.

12) The authors should give more explanations regarding their choices to feed the conceptual mathematical model: why choosing a mean of eight molecules for BisDC and TciR (is there a change if this value is changed)? Does it rely on particular biological data (level of production of proteins?)? Why choosing this particular binomial distribution for the other proteins? What does "bin size=1" (indicated on the figure) mean? What does the bracket indicate on panel 2? Why not feeding the model with real biological data in particular affinities of the regulator for the targeted promoters (to get values for A1/binding and A2/unbinding of regulators)?

13) Although key regulators of ICE*clc* have been characterized, the full cascade of regulation is not completely deciphered (as stated in the Discussion): role of AlpA, of RpoS, mechanism of reinforcement present in wild-type configuration of the ICE and not restored by in trans induction of plasmid-clones bisDC. The authors should make it clear to the reader that these are still outstanding issues that require future work.

14) Such regulation appears specific to ICE*clc* (even if such mobile elements can be found in several bacterial genus, not only in *Pseudomonas*). In addition, since the regulation cascade is complex and involves several regulators, the manuscript is quite long and requires considerable effort and concentration for the reader (even for a specialist). The authors should take a fresh look and try to really guide the reader through the steps, be as succinct as possible, and not make the manuscript any longer than it already is, in spite of all these suggestions for clarification.

---

## [Author Response]

Essential revisions:1) Along with the description of the underlined mechanisms controlling this process, a novel transcription factor (BisR) was described. Perhaps the new regulator could be emphasized more in the Abstract. The work is very complete and well-performed. The authors do a nice job of walking the reader through the various genetic manipulations that were needed to draw the conclusions on a complex regulatory system, especially in the first part of the Results.

We highly appreciate the judgement of the reviewers in this point.

Action: We have emphasized the discovery of both new regulatory factors (BisR and BisDC) more in the Abstract, and included a more general statement of their widespread nature.

2) The manuscript could be improved as a publication for eLife if the authors argue more than they do now about the general interest of their work, the possible importance of these elements in pathogens, and respond to the few methodological issues raised. A few particular suggestions are made below.

We thank the reviewers for bringing up this point, although it would be sad to align ‘general interest’ with being identical to pathogenicity and antibiotic resistance. The ICE of this class are extremely widespread as we mention in the second paragraph of the Introduction; and we have previously shown general conservation of the ‘core’ ICE*clc* structure to several elements in pathogenic bacteria, some of which also carry genes considered to be implicated in virulence and antibiotic resistance. Many of the detected elements through genome sequencing have so far remained ‘cryptic’ and it requires considerable detail to ‘extract’ their precise boundaries, given the large variability of the non-conserved ICE regions.

Action: We have included more literature and emphasized the presence of ICE*clc* family elements in pathogenic strains. We have emphasized the presence of ICE*clc*-like elements in pathogenic bacteria in the Abstract. We have also highlighted genes within the ICE variable regions potentially relevant for antibiotic resistance and pathogenicity (see Introduction, second paragraph, subsection “BisDC-elements are widespread in other presumed ICEs” and Discussion, last paragraph).

3) Since the clc element transfers by conjugation much like conjugative plasmids, and several plasmids, like those of the IncF family, also seem to transfer only from a limited number of cells (or at least the rates of transfer are very low), is there any information on whether or not conjugation mediated by some groups of plasmids may also be controlled by such a complex system? Given the broad readership of eLife, it would be helpful to broaden the discussion to horizontal (or at least conjugative) transfer of other genetic elements.

We thank the reviewers for this general observation that, indeed, many conjugative elements transfer at frequencies that suggest that not all cells develop a sort of ‘transfer competence’. Regrettably, there are not enough groups investing in single cell analysis to study such developmental bacterial cell fates. Known regulatory systems of other ICE such as ICESXT have an architecture that would enable bistability cell decisions, which may be at the basis of low conjugation frequencies. We have postulated this in a review on ICE single cell behaviour (Delavat et al., 2017).

Action: We have included this in the Introduction (first paragraph) and in the Discussion (last paragraph).

4) Though the role of RpoS is intriguing, and suggests some kind of stress response. Can more be said about that in this study, even though it was not the focus but seems to be critical?

We have previously demonstrated (Miyazaki et al., 2012) that there is a correlation between individual cells carrying higher RpoS levels and their probability of developing transfer competence. That study discovered that RpoS is interacting at the inrR-promoter, and thus partly responsible for late-downstream promoter activation. On the other hand, that study also showed that RpoS is not ‘essential’, and a low level expression remains in absence of RpoS.

We have not specifically studied the potential implication of RpoS on activation of BisR or alpA, although all our measurements were done on stationary phase cells.

Action: We mention the role of RpoS in the and provide a more general outlook in the eighth paragraph of the Discussion.

5) Subsection “A new regulator BisDC is the last step in the activation cascade”: I did not quite understand at first why complementation of a bisD deletion mutant with bisCD did not result in similar frequencies of transfer as in the wild-type or other deletion mutants. It would be helpful to elaborate a bit more on this concept of 'reinforcement' at this stage in the paper, more specifically how these findings led to that conclusion. Later on, when the positive feedback is demonstrated, it would be helpful to go back and explain this result.

Thanks for pointing this out.

Action: We have elaborated more on this feedback in the subsection “A new regulator BisDC is the last step in the activation cascade”, with the first time results.

6) The only place where I was a bit lost during the first read-through was the evidence of a bistable output (see below). I think the average reader will be puzzled by the fact that you equate bistable with 'digital' and a single signal with 'analog'. Some explanation is warranted here. Moreover, is this the only circuit working as analog-to-digital converter in bacteria? More comparison to other systems in bacteria was missing. Can you elaborate features that seem unique so far in ICEclc and those that are similar to other systems.

We thank the reviewer for pointing this out. Indeed, most phenotypes among cells in a population can be considered as an ‘analog’ behaviour with a global mean and variation around that mean. Bistable decisions in contrast, are a form of digital behaviour, in which single cells follow either one or the other phenotypic fate (‘yes’ or ‘no’). ICE*clc* can thus impinge on, for example, the variation in levels of a transcription factor among all cells in the population, and integrate this to its transfer competence pathway in a subset of cells.

This is very likely not the only analog-to-digital converter in bacteria, because any set of bistable developments creates digital behaviour, as we mentioned in the Discussion.

The importance here is to realize that once a digital conversion is set in motion, that particular cell has to remain in this mode and should not ‘escape’ from its developmental path.

Action: We have rephrased and explained this in more detail in the Abstract, Introduction (last paragraph), the Results (subsection “Modeling suggests positive feedback loop to generate and maintain ICE*clc* bistable output”) and the Discussion (second and seventh paragraphs).

7) In general terms, the authors did not put much emphasis on the role of AlpA on the modulation of the circuit. What is this element? Which could be the potential mechanisms for its effect on PalpA? It is clear this element is relevant for the systems, but this has been only superficially mentioned in the work. It would strengthen the work if this gene would be investigated a bit more, at least by bioinformatics.

AlpA is originally known as a phage repressor in *E. coli*. (There is also an alpA in *P. aeruginosa* (McFarland et al., Proc Natl Acad Sci U S A 112, 8433-8438, 2015), but this is an unrelated transcription regulator). The ‘AlpA’-domain is a well-defined and wide-spread DNA binding domain. Bioinformatics therefore suggests the AlpA of ICE*clc* (and homologs) to be a DNA binding protein, but there is insufficient information to conclude whether all these ‘AlpA’-homologs are activators or repressors, or something else.

Action: We include AlpA in Figure 1—figure supplement 2, together with BisR, BisD and BisC, and describe it in the Results subsection “BisDC is part of a positive autoregulatory feedback loop”.

8) The authors exploit the phylogenetic distribution of ICEclc in several organisms. Have this element (or similar) been described associated with pathogenesis or antibiotic resistance? It was mentioned early at the Introduction the association of ICE elements with antibiotic elements, but has the key regulatory elements of ICEclc been identified in pathogenic bacteria or associated with virulence? This information would enhance the general interest of the work beyond environmental bacteria and biodegradation.

We appreciate the suggestion to reinforce this aspect. We have previously found by genome analysis (Miyazaki et al., 2011 and 2014) that ICE similar to ICE*clc* occur in several opportunistic pathogens, such as *P. aeruginosa*, *Bordetella petri* and *B. bronchiseptica*, as well as the plant pathogens *Xylella fastidiosa* and *Xanthomonas campestris*. That study mentioned occurrence of antibiotic resistance determinants on several of those ICEs. Older studies (e.g., Klockgether et al., 2007) have shown the similarities between ICE*clc* and pathogenicity islands of the type PAGI-2 and PAGI-3 in *P. aeruginosa* clinical isolates.

Action: We have included this information in the Introduction (third paragraph), and have further searched for gene functions of potential pathogenic character or antibiotic resistance among the list of newer ICEs mentioned in Figure 1—figure supplement 3. Importantly we found a newer study on elements named PAGI-16 in hundreds of clinical *P. aeruginosa* isolates that have the same core structure as ICE*clc* but carry carbapenem resistance. This information was included in the subsection “BisDC-elements are widespread in other presumed ICEs” and the last paragraph of the Discussion.

9) Why do the authors use 75th of relative fluorescence? What is the rational for that? What would happen if relative fluorescence for all cells were used?

We apologize for only referring to previous literature here and not explaining this more extensively. The use of the 75^th^ percentile instead of the median or mean is justified when distributions of values are extremely skewed, as is the case for subpopulation-dependent expression that characterizes many of the observed reporter expressions here (see, for example, Figure 4—figure supplement 1 and 2, violin plots). This aspect has been treated in detail with simulations in Gaillard et al., 2010. For bistable expression, the only proper way to estimate the size and expression levels of both subpopulations is to use QQ-plots, although 75^th^ and 95^th^ percentiles have been used before. Use of a simple ‘mean’ or ‘median’ is insufficient and the second ‘bistable’ population cannot be discerned.

Action: We have explained this in more detail in the Materials and methods section.

10) The authors used different single-copy reporter systems to investigate promoter activity. For miniTn5, 3 independent clones were used. Yet, even in this case the system will not be isogenic. It would be better to have all reporter systems using the same insertion locus, such as miniTn7, to have a faithful composition between strains. In terms of methodology, for me it’s is the only concern that I have.

This is a correct remark and we do find occasionally higher than expected differences among the independent clones of Tn5 insertions (although not here, see, for example Figure 7C individual replicate data). However, for most reporters described here we actually did use mini-Tn7 (see Table 1), and the only reason to use mini-Tn5 insertions was to be able to combine various single-copy components, such as single copy bisR plus a single copy reporter.

Action: We explained this in the Materials and methods subsection “Reporter gene constructs”.

11) One concern is that the authors used two approaches to decipher the regulatory cascade of this ICE. The first one relied on the mutagenesis of putative regulator genes of ICEclc and complementation by genes cloned in plasmids. The second one consisted in the ectopic expression of individual and combinations of suspected regulatory elements in a host without ICEclc and study of the expression of single chromosomal copy of transcriptional fusions. Both methods rely on ectopic expression of the regulator genes and thus in an "off-ground" analysis, i.e. not in the in situ context of transcription of ICE genes. Thus potential regulation elements can be missed: secondary structures of long RNA transcripts, competition between regulators, dosage of regulators versus promoters etc.

We appreciate this concern, but we think it is not valid here, because the epistasis experiments were conducted in background where wild-type or mutant ICE*clc* were present. See, for example, Figures 2C, 3C, 4C. Therefore, any further potential regulation elements were present. We note this in the ninth paragraph of the Discussion.

Action: no further action needed.

12) The authors should give more explanations regarding their choices to feed the conceptual mathematical model: why choosing a mean of eight molecules for BisDC and TciR (is there a change if this value is changed)? Does it rely on particular biological data (level of production of proteins?)? Why choosing this particular binomial distribution for the other proteins? What does "bin size=1" (indicated on the figure) mean? What does the bracket indicate on panel 2? Why not feeding the model with real biological data in particular affinities of the regulator for the targeted promoters (to get values for A1/binding and A2/unbinding of regulators)?

We thank the reviewer for these remarks. The importance of the model is primarily conceptual, to show how bistability can arise and pertain in particular configurations. We currently have no biochemical data of binding and unbinding constants, and this may be very difficult given that these proteins have not been purified.

However, we assume that the regulatory proteins oligomerize similarly as is known from many systems (e.g., tetramers for LysR members, dimers or heteromers). This is mentioned in the legend to Figure 6, with reference to a global overview of transcription factors (Tropel, 2004).

The mean assumed cellular levels of regulatory protein concentrations were based on measurements of similar regulatory proteins in e.g., *E. coli* (new reference included; Li et al., 2014). We varied our simulations in the range of these values, as e.g., shown in panels C and D, and showed the expected outcomes.

We assumed that such regulatory proteins as TciR would be occurring in all cells in a population, with per cell levels uniformly distributed, to show the outcome on bistability in the circuit.

Bin size referred to the shown distributions, but this was removed for clarity.

The bracket referred to the integration of protein levels after 100 time step and across 10,000 simulations.

Action: we improved Figure 6 and its legend with the elements mentioned here. We detailed the text referring to this figure in the subsection “Modeling suggests positive feedback loop to generate and maintain ICE*clc* bistable output”.

13) Although key regulators of ICEclc have been characterized, the full cascade of regulation is not completely deciphered (as stated in the Discussion): role of AlpA, of RpoS, mechanism of reinforcement present in wild-type configuration of the ICE and not restored by in trans induction of plasmid-clones bisDC. The authors should make it clear to the reader that these are still outstanding issues that require future work.

We appreciate this remark. We have indicated this ourselves in the ninth paragraph of the Discussion. However, our experimental results indicate that the main elements that cause bistability have been captured by our study. They are sufficient by themselves to produce bistability (Figure 7) in absence of other ICE functions.

Action: We have modified the Discussion to indicate the various outstanding issues (sixth paragraph). We have split Figure 1B into the previous model of regulation and Figure 1—figure supplement 1 that shows the new updated model, and the further outstanding questions.

14) Such regulation appears specific to ICEclc (even if such mobile elements can be found in several bacterial genus, not only in Pseudomonas). In addition, since the regulation cascade is complex and involves several regulators, the manuscript is quite long and requires considerable effort and concentration for the reader (even for a specialist). The authors should take a fresh look and try to really guide the reader through the steps, be as succinct as possible, and not make the manuscript any longer than it already is, in spite of all these suggestions for clarification.

Point well taken. We understand that the matter is complex in its presentation.

Action: We have combed through the complete text to be as succinct as possible, and to guide readers through the text, figures and legends.